



# Time of Emergence of compound events: contribution of univariate and dependence properties

Bastien François[1] and Mathieu Vrac[1]

[1]Laboratoire des Sciences du Climat et l'Environnement (LSCE-IPSL) CNRS/CEA/UVSQ, UMR8212, Université Paris-Saclay, Gif-sur-Yvette, France

**Correspondence:** B. François (bastien.francois@lsce.ipsl.fr)

**Abstract.**

Many climate-related disasters often result from a combination of several climate phenomena, also referred to as "compound events" (CEs). By interacting with each other, these phenomena can lead to huge environmental and societal impacts, at a scale potentially far greater than any of these climate events could have caused separately. Marginal and dependence properties of the climate phenomena forming the CEs are key statistical properties characterising their probabilities of occurrence. In this

study, we propose a new methodology to assess the time of emergence of compound events probabilities, which is critical for mitigation strategies and adaptation planning. Using copula theory, we separate and quantify the contribution of marginal and dependence properties to the overall probability changes of multivariate hazards leading to compound events. It provides a better understanding of how the statistical properties of variables leading to CEs evolve and contribute to the change of their

occurrences. For illustration purposes, the methodology is applied over a 13-member multi-model ensemble (CMIP6) to two case studies: compound wind and precipitation extremes over the region of Brittany (France), and frost events occurring during the growing season preconditioned by warm temperatures (growing-period frost) over Central France. For compound wind and precipitation extremes, results show that probabilities emerge before the end of the 21st century for 6 models of the considered CMIP6 ensemble. For growing-period frosts, significant changes of probability are detected for 11 models. Yet, the contribution

of marginal and dependence properties to these changes of probabilities can be very different from a climate hazard to another, and from one model to another. Depending on the CE, some models give a strong importance to both marginal properties and dependence properties for probability changes. These results highlight the importance of considering both marginal and dependence properties changes, as well as their inter-model variability, for future risk assessments due to compound events.



## 1   Introduction

In September 2017, heavy rainfall and storm surge associated with Hurricane Irma resulted in record-breaking floods in Jacksonville, Florida. In 2019, Australia had experienced high temperatures and prolonged dry conditions, which resulted in one of the worst bush fire seasons in its recorded history. In April 2021 and 2022, Central Europe experienced consecutive days of frost events following a warm early spring, which caused severe damages to agricultural yields. These recent climate events are

some examples of so-called "compound events" (CEs), i.e., high-impact climate events that result from interactions of several climate hazards. These climate hazards are not necessarily extremes themselves, but their simultaneous or successive occurrences can generate strong impacts (Leonard et al., 2014; Zscheischler et al., 2014, 2018, 2020). Though still in its infancy, the understanding of the complex nature of compound events and the assessment of their associated risks have been the subject of numerous research studies in climate sciences (e.g., Bevacqua et al., 2017, 2021; Manning et al., 2018; Zscheischler and

Seneviratne, 2017; Ridder et al., 2021, 2022; Singh et al., 2021a; Nasr et al., 2021; Raymond et al., 2022, among many others). Recently, a typology of compound events has been proposed in order to categorise them into four classes depending on how individual hazards interact to form the CEs ("preconditioned", "multivariate", "temporally compounding" and "spatially compounding" events, see Zscheischler et al., 2020). Concerning projected changes, frequency and intensity of some compound events such as co-occurring heatwaves and droughts are expected to increase for many regions of the world, even when consid-

ering climate change scenarios with limited global warming to 1.5°C above pre-industrial levels (IPCC, 2021). Determining whether probabilities of compounding climate events present significant changes between past and future periods, and to detect when these significant changes occur are of paramount importance, not only for mitigation and adaptation issues but also to inform the general public and to raise awareness of climate change. Only when the changes of probability are of sufficient magnitude relative to a baseline period can we be confident that significant changes have been detected. Detecting from which

period the changes are statistically significant corresponds to the concept of "Time of Emergence" (ToE). It consists in determining the time or period in which a climate signal emerges from (i.e., goes out of) the natural variability (e.g., Christensen et al., 2007; Maraun, 2013; Hawkins et al., 2020; Ossó et al., 2022). Time of Emergence has been discussed extensively to analyse the emergence of mean temperatures (e.g., Hawkins and Sutton, 2012; Mahlstein et al., 2011), precipitation (Fischer et al., 2014; Giorgi and Bi, 2009; Gaetani et al., 2020), but also emergence of extremes (e.g. Diffenbaugh and Scherer, 2011;

Fischer et al., 2014; King et al., 2015). Evaluating the ToE of compound hazards probabilities with respect to a baseline period — from which the natural variability is estimated — is valuable to analyse evolutions of compound events and attribute those to a specific cause, such as anthropogenic greenhouse gas emissions. Attribution is an important research field in climate science that aims at determining the mechanisms responsible for recent global warming and related climate changes. For example, it can be done by comparing probabilities of an event between two worlds with different forcings (the "risk-based" approach,

Stott et al., 2004; Shepherd, 2016). Generally, a factual world with anthropogenic climate change and a counterfactual world in which anthropogenic emissions had never occurred are considered. Although we do not aim at performing attribution per se in the present study, the underlying philosophy is relatively similar for ToE: by considering a pre-industrial period as baseline,



compound hazards probabilities associated with natural forcings — or natural variability — could be estimated, and so the influence of future climate change on probabilities.

From a statistical point-of-view, compound events are characterised by the statistical features of the variables forming the CEs, i.e., their marginal properties (e.g., mean and variance) and dependence structures. These key statistical properties can be affected by future climate change (e.g., Wahl et al., 2015; Schär, 2015; Russo et al., 2017; Raymond et al., 2020; Jézéquel et al., 2020). In addition to potentially exacerbate impacts, these evolutions of marginal and dependence properties could also combine to change the probabilities of the CEs' hazards (e.g., Rana et al., 2017; Zscheischler and Seneviratne, 2017; Zscheischler

and Lehner, 2021; Manning et al., 2019; Singh et al., 2021a). For example, rising temperatures can naturally lead to more co-occurrences of hot temperatures and droughts, despite no significant trends in droughts are detected (Diffenbaugh et al., 2015; Mazdiyasni and AghaKouchak, 2015). However, in addition to warmer temperatures, the strengthening of the dependence between hot temperatures and droughts for future periods can also contribute to an increase in their co-occurrences (as highlighted in Zscheischler and Seneviratne, 2017). Several studies concluded about the importance of considering dependencies to

assess CE properties and frequencies in a robust way (e.g., Hillier et al., 2020; Singh et al., 2021a; Vrac et al., 2021). Recently, Abatzoglou et al. (2020) even showed, using reanalysis data, that changes in dependence properties have been more important than changes in univariate properties in the recent decades. Hence, to determine the ToE of hazards probabilities, quantifying the influence (or contribution) of the statistical features of the variables forming the CEs to these changes of probabilities is thus crucial to further understand the potential future evolutions of compound events (Vrac et al., 2021).

In this paper, we propose a new methodology to assess the time of emergence of compound events probabilities. We also develop a copula-based multivariate framework, which allows for an adequate description of the contribution of the marginal and dependence properties changes to the evolutions of multivariate hazard probabilities. This compound event analysis is applied to two case studies. We first analyse compound wind and precipitation extremes over the coastal region of Brittany (France). This bivariate compound event, i.e., composed of co-occurring climate hazards over the same region and time,

has been analysed in several studies (e.g., Martius et al., 2016; Bevacqua et al., 2019; De Luca et al., 2020a; Reinert et al., 2021; Messmer and Simmonds, 2021) as it can have severe impacts such as important economic losses, massive damages to infrastructure and loss of human life (e.g., Fink et al., 2009; Liberato, 2014; Wahl et al., 2015; Raveh-Rubin and Wernli, 2015). We then apply our methodology to a second climate hazard: frost events occurring during the growing season preconditioned by warm temperatures (growing-period frost) over Central France. When occurring after bud burst, i.e., when the sensitive emerging leaves and flowers have started to develop, frost temperatures potentially affect growth and distribution limits of

plants. It can consequently cause important economic losses to agriculture (Lamichhane, 2021). These growing-period frost events and their associated risks in past and future periods have been studied in the literature (e.g., Unterberger et al., 2018; Liu et al., 2018a; Sgubin et al., 2018; Pfleiderer et al., 2019), as well as the role of human-caused climate change on growing-period frosts probability (Vautard et al., 2021).

The rest of this paper is organised as follows: Sect. 2 describes the climate simulations used in this study, and Sect. 3 details the statistical method and experimental setup used to analyse time of emergence of compound events probabilities and contributions of the statistical features. Then, results for the analysis of the two climate compound hazards are provided in



Sect. 4 for extremes of wind and precipitation and in Sect. 5 for growing-period frost events. Conclusions, discussions and perspectives for future research are finally proposed in Sect. 6.

## 2   Model data

One ensemble of 13 Global Climate Models (GCMs) following the CMIP6 protocol (Eyring et al., 2016) is considered. This selection of models is listed in Table 1. To define compound wind and precipitation extremes, we use daily precipitation and wind speed maxima variables. For growing-period frosts, mean and minimum temperature variables are used. For each variable, the historical period simulations (1871-2014) have been extracted and extended until 2100 using the shared socioeconomic pathways 585 (SSP-585) scenario (Riahi et al., 2017). As the 13 selected simulations present different spatial resolutions, each climate simulation dataset has been regridded to a common spatial resolution of $0.5° \times 0.5°$ using bilinear interpolation. Considering the climate models separately will allow us to assess inter-model variability in terms of time of emergence of compound events probabilities, as well as the potentially different contributions of marginal and dependence properties to changes in probability of multivariate climate hazards. Also, by considering all climate models together using a pooling procedure, a multi-model ensemble estimate for ToE and contributions could be derived. Pooling the models together will allow us to better take into account the global uncertainty inherent in climate modelling and to reduce the influence of natural variability amongst individual ensemble members.

For compounding wind and precipitation extremes, we use spatial mean of daily wind speed maxima and spatial sum of daily precipitation time series during winter (December, January and February) over the region of Brittany, France ([-5, -2°E] × [46.5, 49°N], see Fig. 1a), which corresponds to a domain with 21 continental grid cells in our regridded climate simulations. This coastal region is regularly impacted by mid-latitude extra-tropical storms causing large damages to infrastructures (e.g., the storm Xynthia in 2010). Analysing the evolution of probability of compound wind and precipitations extremes is therefore relevant for this region. To allow for a robust statistical modelling of compounding wind and precipitation extremes, we applied our methodology to bivariate points of high values by selecting wind and precipitation data concurrently exceeding selected high thresholds. Indeed, our methodology detailed later in Sect. 3 is based on the use of parametric models and considering the complete bivariate distribution to fit marginals and copulas could be not appropriate as the representation of the extremes would be biased by the bulk of the bivariate distributions where most of the data is located (e.g., Bevacqua et al., 2019). More details on selection thresholds will be provided later in Sect. 4.

For growing-period frost events, data are extracted over Central France ([-1, 5°E] × [46, 49°N], see Fig. 1a), which corresponds to 78 continental grid cells. The region covers an important agriculture area of France, including grapevine and fruit crops with high production (Vautard et al., 2021). We focus on spatial mean of daily minimum temperature ($T$) in April to define frost events occurring in early spring. To account for phenology and characterise bud burst conditions by the end of March, the Growing Degree Day (GDD) model (Bonhomme, 2000) is used. The GDD model consists in computing cumulative daily mean temperatures minus a "base temperature" from a starting date. For our study, a base temperature of 5°C is used and the starting date for computing GDD values for each year is chosen to be 1 January. In this study, our aim is not to focus on the





phenology of specific plants but rather to provide a general overview of growing-period frost events. 5°C as base temperature is generally accepted for crops and grapevine (e.g., Skaugen and Tveito, 2004; Jiang et al., 2011; Ruosteenoja et al., 2016; Vautard et al., 2021). Bud burst occurs when the cumulative sum of degree-days up to 31 March is larger than some thresholds (Garcia de Cortazar-Atauri et al., 2009), which depend on species. For each year $y$, GDD values by the end of March are obtained via the formula:

$$GDD(y) := \sum_{i=y/01/01}^{i=y/03/31} \max(MT(i) - 5, 0),$$

with $MT$ the daily mean temperature. GDD values are computed for each grid cell and averaged spatially over the area of Central France. We consider the threshold of 200°C.day to characterise bud burst conditions and illustrate our method. The choice of this threshold is consistent with existing studies analysing bud burst values of grapevine species (e.g., Garcia de Cortazar-Atauri et al., 2009; Vautard et al., 2021), and is useful to characterise early bud burst plants that could be impacted by frost events.

For illustration purpose, Fig. 1a displays the topographic map of France with the region of Brittany and Central France in boxes. The bivariate wind and precipitation data (Fig. 1b) and miniminal temperature and GDD data (Fig. 1c) for the CNRM-CM6 model are also displayed.

## 3 Statistical method

Our aim is to design a statistical method to assess the time of emergence of compound events probabilities, that is, to detect from which period changes of probability are statistically significant relative to a baseline period. Probabilities of compound events can be computed with copulas. Copulas are functions that allow to describe the dependence structure between random variables separately from their marginal distributions and greatly simplifies calculations involving multivariate distributions (Nelsen, 2006). Copulas have been widely applied in climate and geophysical science (e.g., Vrac et al., 2005; Salvadori et al., 2007; Schölzel and Friederichs, 2008; Serinaldi, 2014). In addition to allowing computations of multivariate hazards probabilities, the use of copulas in our study permits to isolate and quantify the marginal and dependence contributions of the variables forming the CEs to the overall probability changes. In the following, we first remind the concept of ToE, and then present our methodology to assess the time of emergence of compound events probabilities. Then, after some reminders about the copula theory, the methodology to assess the contribution of marginal and dependence properties to changes of probabilities is presented. For ease of presentation, the methodology is explained for compounding wind and precipitation extremes but will be applied similarly for growing-period frosts.

### 3.1 Time of emergence of climate hazards

The concept of Time of Emergence (ToE) has been developed to assess the significance of climate changes relative to background variability. Comparing changes of climate signal relative to the natural variability is particularly relevant as human





societies and ecosystems are inherently adapted to the local background level of variability, and major impacts arise most likely when changes emerge from it (e.g., Lobell and Burke, 2008). Different methodologies to assess ToE of climate signals have been used in the literature. For example, ToE can be assessed by estimating the climate change signal (S) and the variability (or noise, N) of the climate metric of interest (e.g., Hawkins and Sutton, 2012; Maraun, 2013; Hawkins et al., 2020; Ossó et al., 2022). The ToE is then estimated by determining the first period for which the S/N ratio permanently crosses a certain threshold (e.g., emergence of "unusual" (S/N > 1), "unfamiliar" (S/N > 2), or "unknown" (S/N > 3) climates, Frame et al., 2017). Methodologies for ToE based on statistical tests have also been developed, which estimate the first period for which the distribution of the climate metric is significantly and permanently different from a baseline period distribution (e.g. using Kolmogorov-Smirnov tests, Mahlstein et al., 2012; Gaetani et al., 2020; Pohl et al., 2020). To define emergence of compound events probabilities, we propose to assess probabilities in a 30-year window sliding over the period 1871-2100 and compare their values with respect to a baseline period's probability. In this study, we consider the reference period (1871-1900) as baseline to assess the emergence of hazard probabilities. However, there is no agreement on the choice of the baseline period for ToE studies. While most of the studies choose a pre-industrial period as baseline to attribute emergence to anthropogenic greenhouse gas forcing (e.g., 1850-1900, Hawkins et al., 2020), other studies choose a more recent baseline period (e.g., 1951-1983, Ossó et al., 2022), which can provide relevant information for adaptation planning. We further discuss the choice of the reference period for emergence in Sect. 6. The ToE of hazard probabilities is then the time period when a significant change of probability occurs relatively to the probability associated with the estimated natural variability, and persists until the end of the century. To assess if probabilities are significantly different from that of the background variability, we propose to compute the 68% and 95% confidence intervals of the baseline period's probability. It permits to characterise the natural variability of our probability of interest. An emergence is detected if probability for the 30-year sliding windows permanently go out of the baseline confidence intervals (i.e., out of the estimated natural variability). The ToE is then defined as the central year of the sliding window over which the probability starts to emerge. As probabilities are estimated using copula modelling (see later in subsection 3.2), 68% and 95% confidence intervals of baseline period's probabilities are computed by coupling the parameters uncertainties of both the fitted marginal distributions and the fitted copula. Considering both 68% and 95% confidence intervals allows to evaluate, with different degrees of confidence, the changes of probability of compounding events from the estimated natural variability. Details on the procedure to compute confidence intervals are given in Appendix A.

## 3.2 A reminder on copula functions and exceedance probability

In this study, we use copula modelling to compute compound events probabilities. We first consider two random variables $X$ (e.g., maximum wind speed) and $Y$ (e.g., precipitation) for an arbitrary period. We denote their marginal (i.e., univariate) probability density functions (pdfs) $f_X(x)$ and $f_Y(z)$ and cumulative marginal distribution functions (CDFs) $F_X(x) = \mathbb{P}(X \leq x)$ and $F_Y(y) = \mathbb{P}(Y \leq y)$. Sklar's theorem (Sklar, 1959) states that, $H$, the joint (i.e., bivariate) CDF can be written as:

$$H_{X,Y}(x,y) = \mathbb{P}(X \leq x \cap Y \leq y) = C(F_X(x), F_Y(y)), \tag{1}$$




where $C$ is a function called "copula", corresponding to the joint distribution function of the uniformly distributed variables $F_X(X)$ and $F_Y(Y)$. Under the assumption that the marginal distributions $F_X$ and $F_Y$ are continuous, Sklar's theorem states that the copula $C$ is unique. This decomposition of the multivariate distribution into marginals distributions and copula function allows us to model the dependence among contributing variables independently of their marginals. Therefore, using copulas makes it easy to isolate the effects of marginal and dependence properties on probability of multivariate hazards.

Bivariate exceedance probability refers to the probability that both random variables exceed a certain value ("AND approach", Salvadori et al., 2016) and can be calculated relatively easily using copulas. For example, for wind and precipitation compound events, it corresponds to probabilities of wind speed and precipitation jointly exceeding established thresholds . We denote $p_{m,d}$ the bivariate exceedance probability computed with marginal (subscript $m$) and dependence (subscript $d$) properties of $(X, Y)$. The probability $p_{m,d}(t_X, t_Y)$ that both $X$ and $Y$ jointly exceed some predefined thresholds $t_X$ and $t_Y$ is given by (Yue and Rasmussen, 2002; Shiau, 2003):

$$p_{m,d}(t_X, t_Y) = \mathbb{P}(X \geq t_X \cap Y \geq t_Y)$$
$$= 1 - F_X(t_X) - F_Y(t_Y) + C(F_X(t_X), F_Y(t_Y)). \tag{2}$$

Marginal and copula distributions in Eq. (2) are estimated using parametric fitting procedures. More details on the fitting procedures for compound wind and precipitation extreme and growing-period frost events are given in Appendix B.

### 3.3 Change of probabilities: contribution of the marginal and dependence properties

Let us now consider the realizations $(X_{\text{ref}}, Y_{\text{ref}})$ and $(X_{\text{fut}}, Y_{\text{fut}})$ of the two random variables $X$ and $Y$ over the reference period (i.e., 1871-1900 in the following), and over another 30-year period (e.g. a future period such as 2071-2100). Using Eq. (2), the reference and future bivariate exceedance probability $p_{m_{\text{ref}}, d_{\text{ref}}}(t_X, t_Y)$ and $p_{m_{\text{fut}}, d_{\text{fut}}}(t_X, t_Y)$ for some predefined thresholds $t_X$ and $t_Y$ are given by:

$$p_{m_{\text{ref}}, d_{\text{ref}}}(t_X, t_Y) = 1 - F_{X_{\text{ref}}}(t_X) - F_{Y_{\text{ref}}}(t_Y) + C_{\text{ref}}(F_{X_{\text{ref}}}(t_X), F_{Y_{\text{ref}}}(t_Y)), \tag{3}$$

$$p_{m_{\text{fut}}, d_{\text{fut}}}(t_X, t_Y) = 1 - F_{X_{\text{fut}}}(t_X) - F_{Y_{\text{fut}}}(t_Y) + C_{\text{fut}}(F_{X_{\text{fut}}}(t_X), F_{Y_{\text{fut}}}(t_Y)). \tag{4}$$

As modeled here with Eqs. (3) and (4), $p_{m_{\text{fut}}, d_{\text{fut}}}$ and $p_{m_{\text{ref}}, d_{\text{ref}}}$ can differ due to:

– changes in the marginal properties of $X$ and $Y$, i.e., changes between $F_{X_{\text{ref}}}$ and $F_{X_{\text{fut}}}$, as well as between $F_{Y_{\text{ref}}}$ and $F_{Y_{\text{fut}}}$,

– and changes in the dependence structure (i.e., in the copulas) between $X$ and $Y$, i.e., changes between $C_{\text{ref}}$ and $C_{\text{fut}}$.

Then, do exceedance probability values change significantly between reference and future periods? And if so, how much of this change is due to changing marginal properties? To changing dependence structure? In order to isolate the effects of



these potentially changing statistical properties, we propose to calculate two additional exceedance probability values. The
first one is the probability $p_{m_{\text{fut}}, d_{\text{ref}}}$, which assesses what the future probability would be if only the marginal properties change
between the reference and future period (and thus keeping the dependence properties from the reference period). $p_{m_{\text{fut}}, d_{\text{ref}}}$ is
hence computed as:

$$p_{m_{\text{fut}}, d_{\text{ref}}}(t_X, t_Y) = 1 - F_{X_{\text{fut}}}(t_X) - F_{Y_{\text{fut}}}(t_Y) + C_{\text{ref}}(F_{X_{\text{fut}}}(t_X), F_{Y_{\text{fut}}}(t_Y)). \tag{5}$$

Inversely, the second additional probability $p_{m_{\text{ref}}, d_{\text{fut}}}$ is aimed to assess what the future probability would be if only the
dependence properties change between the reference and future period (keeping the marginal properties from the reference
period), and is computed as:

$$p_{m_{\text{ref}}, d_{\text{fut}}}(t_X, t_Y) = 1 - F_{X_{\text{ref}}}(t_X) - F_{Y_{\text{ref}}}(t_Y) + C_{\text{fut}}(F_{X_{\text{ref}}}(t_X), F_{Y_{\text{ref}}}(t_Y)). \tag{6}$$

Illustrations of these four probabilities for artificial bivariate distributions and changes between a reference and a future period
are given in Fig. 2.

To assess how much marginal and dependence contribute to exceedance probabilities change between reference and future
period, we use the four probabilities derived above to decompose the overall probability change. We first define $\Delta\text{P}$, the change
of probability between the reference and future periods, as the difference between the two probabilities: $\Delta\text{P} = p_{m_{\text{fut}}, d_{\text{fut}}} - p_{m_{\text{ref}}, d_{\text{ref}}}$. By computing $p_{m_{\text{fut}}, d_{\text{ref}}}$ and $p_{m_{\text{ref}}, d_{\text{fut}}}$, one can decompose the change of probability $\Delta\text{P}$ into a sum of three terms that
can yield statistical interpretations:

$$\Delta\text{P} = \Delta\text{M} + \Delta\text{D} + \Delta\text{I}. \tag{7}$$

The first term $\Delta\text{M}$ accounts for the difference of probability between the reference and future periods due to a change of
marginal properties only and is hence called the "marginal" term:

$$\Delta\text{M} = p_{m_{\text{fut}}, d_{\text{ref}}} - p_{m_{\text{ref}}, d_{\text{ref}}}$$

Similarly, the second term $\Delta\text{D}$ assesses the difference of probability between the reference and future periods due to a change
of dependence properties only and is hence called the "dependence" term:

$$\Delta\text{D} = p_{m_{\text{ref}}, d_{\text{fut}}} - p_{m_{\text{ref}}, d_{\text{ref}}}$$

As simultaneous changes of marginal and dependence properties between the reference and future period can affect the
exceedance probability in a highly non-linear fashion (as it can be observed visually in Fig. 2), $\Delta\text{P}$ cannot be simply expressed
as the sum of the differences $\Delta\text{M}$ and $\Delta\text{D}$. Thus, a residual term $\Delta\text{I}$, called the "interaction" term, is introduced to assess the
part of the probability change that is due to the simultaneous change of marginal and dependence properties and that cannot be
explained by the changes of these statistical properties separately:

$$\Delta\text{I} = p_{m_{\text{fut}}, d_{\text{fut}}} - p_{m_{\text{fut}}, d_{\text{ref}}} - p_{m_{\text{ref}}, d_{\text{fut}}} + p_{m_{\text{ref}}, d_{\text{ref}}}.$$





The decomposition of $\Delta$P into these three terms allows to isolate the effects of the changes of marginal, the effects of the changes of dependence properties and the effects of the changes of interaction on the overall change of probability value $\Delta$P.

By taking advantage of this decomposition, we propose to quantify the contribution (in %) of the different terms $\Delta$M, $\Delta$D and $\Delta$I to the change of probability $\Delta$P. For example, the contribution of the changes of the marginal properties can be quantified as:

$$\mathrm{Contrib}_{\Delta M} = \frac{\Delta M}{\Delta P} \times 100, = \frac{p_{m_{\mathrm{fut}},d_{\mathrm{ref}}} - p_{m_{\mathrm{ref}},d_{\mathrm{ref}}}}{p_{m_{\mathrm{fut}},d_{\mathrm{fut}}} - p_{m_{\mathrm{ref}},d_{\mathrm{ref}}}} \times 100. \tag{8}$$

A value of 50 % for $\mathrm{Contrib}_{\Delta M}$ would indicate that the change of marginal properties is responsible for 50 % of the global

change of probability $\Delta$P between the reference and future periods. The contributions of $\Delta$D (resp. $\Delta$I) can be calculated the same way by simply replacing $\Delta$M in Eq. (8) by $\Delta$D (resp. $\Delta$I). The sum of the three contributions adds up to 100 %, by construction. Please note that, for illustration, changes of probability $\Delta$P, $\Delta$M and $\Delta$D are here considered as differences of probabilities. One could also consider analysing other metrics such as relative differences ("r. diff") by dividing each of the terms in Eq. (7) by $p_{m_{\mathrm{ref}},d_{\mathrm{ref}}}$:

$$
\begin{aligned}
\quad \Delta \mathrm{P}^{\text{r. diff}} &= \frac{p_{m_{\mathrm{fut}},d_{\mathrm{fut}}} - p_{m_{\mathrm{ref}},d_{\mathrm{ref}}}}{p_{m_{\mathrm{ref}},d_{\mathrm{ref}}}}, \\
\Delta \mathrm{M}^{\text{r. diff}} &= \frac{p_{m_{\mathrm{fut}},d_{\mathrm{ref}}} - p_{m_{\mathrm{ref}},d_{\mathrm{ref}}}}{p_{m_{\mathrm{ref}},d_{\mathrm{ref}}}}, \\
\Delta \mathrm{D}^{\text{r. diff}} &= \frac{p_{m_{\mathrm{ref}},d_{\mathrm{fut}}} - p_{m_{\mathrm{ref}},d_{\mathrm{ref}}}}{p_{m_{\mathrm{ref}},d_{\mathrm{ref}}}}, \\
\Delta \mathrm{I}^{\text{r. diff}} &= \frac{p_{m_{\mathrm{fut}},d_{\mathrm{fut}}} - p_{m_{\mathrm{fut}},d_{\mathrm{ref}}} - p_{m_{\mathrm{ref}},d_{\mathrm{fut}}} + p_{m_{\mathrm{ref}},d_{\mathrm{ref}}}}{p_{m_{\mathrm{ref}},d_{\mathrm{ref}}}}.
\end{aligned}
$$

In addition, bivariate fraction of attributable risk ("FAR", e.g., Stott et al., 2016; Chiang et al., 2021; Zscheischler and Lehner, 2021) can also be computed by dividing each of the term by $p_{m_{\mathrm{fut}},d_{\mathrm{fut}}}$:

$$
\begin{aligned}
\Delta \mathrm{P}^{\mathrm{FAR}} &= \frac{p_{m_{\mathrm{fut}},d_{\mathrm{fut}}} - p_{m_{\mathrm{ref}},d_{\mathrm{ref}}}}{p_{m_{\mathrm{fut}},d_{\mathrm{fut}}}}, \\
\Delta \mathrm{M}^{\mathrm{FAR}} &= \frac{p_{m_{\mathrm{fut}},d_{\mathrm{ref}}} - p_{m_{\mathrm{ref}},d_{\mathrm{ref}}}}{p_{m_{\mathrm{fut}},d_{\mathrm{fut}}}}, \\
\Delta \mathrm{D}^{\mathrm{FAR}} &= \frac{p_{m_{\mathrm{ref}},d_{\mathrm{fut}}} - p_{m_{\mathrm{ref}},d_{\mathrm{ref}}}}{p_{m_{\mathrm{fut}},d_{\mathrm{fut}}}}, \\
\quad \Delta \mathrm{I}^{\mathrm{FAR}} &= \frac{p_{m_{\mathrm{fut}},d_{\mathrm{fut}}} - p_{m_{\mathrm{fut}},d_{\mathrm{ref}}} - p_{m_{\mathrm{ref}},d_{\mathrm{fut}}} + p_{m_{\mathrm{ref}},d_{\mathrm{ref}}}}{p_{m_{\mathrm{fut}},d_{\mathrm{fut}}}}.
\end{aligned}
$$

However, by construction, results for contributions, either for relative differences or bivariate FAR, would be identical to those obtained for differences.



## 3.4 Application to the multi-model ensemble

The methodology described above to assess time of emergence of compound events probabilities and marginal and dependence contributions to these changes is applied to the 13 CMIP6 models by considering successively all 30-year sliding windows spanning the period 1871-2100. Moreover, the methodology is applied to the ensemble in two different versions:

- the "Indiv-Ensemble" version, for which the methodology is applied to each climate model individually. In particular for contributions and ToE, multi-model median estimates are derived to summarise the information given by all the models.

- the "Full-Ensemble" version, which consists of pooling the contributing variables of the 13 climate models together and applying the methodology to these pooled data to derive a pooled estimate of time of emergence, as well as marginal and dependence contributions.

Depending on the versions, the objectives are not the same: whereas the Indiv-Ensemble version permits to analyse the modelling of hazards separately and assess the uncertainty in ToE arising from the inter-model differences, the Full-Ensemble version permits to derive unique ToE estimates and contribution values accounting for the global uncertainty in climate modelling. This Full-Ensemble version assumes that the variables of interest are drawn from the same distribution.

Concerning the Full-Ensemble version, a post-processing step of the different models is required for the analysis of compound wind and precipitation extremes only. Indeed, as already explained in Section 2, wind and precipitation data concurrently exceeding high selection thresholds are selected for each climate model in order to focus on compounding extremes. However, climate models can present very different values of wind and precipitation data: for example, a model may not be capable of simulating wind and precipitation events as intense as other models. Hence, each model potentially has different selection thresholds over which values of wind and precipitation are selected. Because of this, selected compound wind and precipitation data from the different climate models cannot be directly pooled, and data need first to be transformed to apply our methodology and analyse pooled extreme events. The transformation step is reached by using a univariate quantile mapping technique (CDF-t, Vrac et al., 2012) that makes the univariate distributions of the wind and precipitation extremes similar to those from a model of reference without modifying their dependence structure. In the following, we choose the CNRM-CM6 model as reference. As values of wind and precipitation extremes of the different models will be modified on purpose by the CDF-t method, note that exceedance thresholds in terms of probabilities (instead of physical values) will be considered. This way, it will enable an interpretation of the results from the Full-Ensemble version. More details about the application of the CDF-t method to transform compound wind and precipitation data for the Full-Ensemble version can be found in Appendix C.

To analyse growing-period frost events with the Full-Ensemble version, no transformation step is needed before pooling. Indeed, contrary to wind and precipitation extreme, the definition of growing-period frost events does not depend on climate models and can be based on well-established thresholds. A summary of the successive steps of our methodology for the Indiv- and Full-Ensemble versions is provided in the form of a flowchart in Fig. 3.




## 4 Results for compounding wind and precipitation extremes

In this section, results are presented for compound wind and precipitation extremes during winter in Brittany. Please note that, for this section as well as for the rest of the study, the period 1871-1900 is considered as the baseline period for natural variability to evaluate time of emergence and contributions. To focus on wind and precipitation extremes, we applied our methodology to points of high values. For each model, we selected points where, concurrently, wind and precipitation values exceed the individual 90th percentiles (denoted $x_{sel}$ and $y_{sel}$, respectively) of the 1871-1900 reference period. In the following, we denote $S^i_{90,90}$ the ensemble of the selected points of high values for a model $i$. For illustration purpose, the ensemble $S^{\text{CNRM-CM6}}_{90,90}$ for the CNRM-CM6 model is shown in orange in Fig 1b. We first illustrate our method with a single climate model (CNRM-CM6). Then, results obtained for the Indiv- and Full-Ensemble versions are presented.

### 4.1 Results for an individual model and a single exceeding threshold: CNRM-CM6

To illustrate our methodology, we first explain the results obtained for compound wind and precipitation extremes and a single bivariate exceeding threshold before extending the results to several bivariate thresholds. We evaluate the probabilities of exceeding the 80th percentiles of the bivariate points belonging to $S^{\text{CNRM-CM6}}_{90,90}$. The 80th percentiles for wind and precipitation correspond to $x_{80|sel} \approx 17.8$ m/s and $y_{80|sel} \approx 338$ mm/d, respectively.

Before computing any probability, Fig. 4 gives a first overview of the fitted bivariate distributions of compound wind and precipitation extremes in our study. It displays the evolutions of the bivariate distributions over a selection of sliding windows due to changing marginal and dependence properties ("Marg.-dep.", Fig. 4a), changing marginal properties only ("Marg.", Fig. 4b) and changing dependence only ("Dep.", Fig. 4c). Plotting these bivariate distributions already indicates the changes in probability of wind and precipitation extremes, and the potential influences of marginal and dependence properties on these changes. Indeed, at first sight in Fig. 4a, the area of bivariate distributions where wind speed and precipitation jointly exceed $x_{80|sel}$ and $y_{80|sel}$ appears to increase for future periods, suggesting that such bivariate events are more likely to occur according to CNRM-CM6 projections. But is this change of probability significant? And is this change due to marginal properties changes? Dependence properties changes? Or both? By keeping the dependence properties of the reference period and considering changing marginal properties only (Fig. 4b), an increase of exceedance probability seems to be observed, although less pronounced. Similar observations can be made by keeping the marginal properties of the reference period and considering changing dependence properties only (Fig. 4c). If both marginal and dependence changes seem to have an importance in the increase of probability, it is important to quantify how much these statistical properties contribute to the change of the overall probability, as well as their respective influence on the time of emergence of probabilities of compounding wind and precipitation extremes.

Time series of exceedance probabilities over all sliding windows for the bivariate threshold ($x_{80|sel}$, $y_{80|sel}$) are presented in Fig. 5 by considering changes of marginal and dependence properties together (Fig. 5a) and separately (Figs. 5b and c). 68% and 95% confidence intervals resulting from marginal and copula uncertainties are also displayed for each probability. All three time series present an increase with time, which is consistent with the visual analysis made in Fig. 4. Probability increase is less





pronounced when future marginal (Fig. 5b) and future dependence properties (Fig. 5c) are considered separately. It illustrates that the effects of these changing statistical properties combine on exceedance probabilities. Yet, all three probability signals permanently go out of the reference natural variability confidence intervals, suggesting that an emergence of probability occurs: for probabilities computed with future marginal and dependence properties (Fig. 5a), the time of emergence is detected in 2009 (1994-2023) and 2072 (2057-2086) for 68 % and 95% confidence levels, respectively. Concerning probabilities influenced by future marginal changes and future dependence changes separately (Figs. 5b and c), probability signals emerge later at the 68 % confidence level, in 2073 (2058-2087) and 2063 (2048-2077), respectively. If contributions of the statistical properties to time of emergence in itself are not computed here, one can get an idea of the importance of the statistical properties on ToE: at the 68% confidence level, ignoring the dependence change would induce a ToE $2073 - 2009 = 64$ years later. Similarly, ignoring marginal changes would induce a ToE $2063 - 2009 = 54$ years later. It thus indicates that both marginal and dependence properties have a non-negligible effect on time of emergence.

Evolution of the bivariate FAR $\Delta \mathrm{P}^{\mathrm{FAR}}$ with respect to the reference period over sliding windows, as well as its decomposition in terms of "marginal" ($\Delta \mathrm{M}^{\mathrm{FAR}}$), "dependence" ($\Delta \mathrm{D}^{\mathrm{FAR}}$) and "interaction" ($\Delta \mathrm{I}^{\mathrm{FAR}}$) terms are displayed in Fig. 5d. As explained in Sect. 3, for each sliding window, the sum of $\Delta \mathrm{M}^{\mathrm{FAR}}$, $\Delta \mathrm{D}^{\mathrm{FAR}}$ and $\Delta \mathrm{I}^{\mathrm{FAR}}$ is by construction equal to $\Delta \mathrm{P}^{\mathrm{FAR}}$. The decomposition highlights that the influences of the marginal and of the dependence properties on bivariate FAR can vary with time. Also, the combination of individual effects of marginal and dependence changes on the overall probability changes is again illustrated: for example, by 2100, considering both future marginal and dependence changes leads to a value of FAR $\Delta \mathrm{P}^{\mathrm{FAR}}$ twice as high as those of $\Delta \mathrm{M}^{\mathrm{FAR}}$ and $\Delta \mathrm{D}^{\mathrm{FAR}}$, respectively. Concerning the interaction term, its associated bivariate FAR is negligible, highlighting that most of the changes can be explained by the changing marginal and dependence properties separately. Results for relative differences are displayed in Fig. 5e, and same conclusions can be drawn. Fig. 5f shows the evolution of the contributions from the marginal, dependence and interaction terms to probability values over sliding windows. By computing the median of contributions over all sliding windows, we can see that both changes in the marginal and in the dependence properties contribute greatly to probability changes ($\approx 50\%$) in the CNRM-CM6 simulations, with a slightly more important contribution from dependence properties (dashed lines in figure 5f). One could remark a symmetry between the contribution values of the marginal and the dependence terms over sliding windows. This can be explained by the way contribution values are computed. Indeed, as the sum of the three contributions adds up to 100 %, by construction, and that the contribution from the interaction term is close to 0, contribution values of the marginal and the dependence terms covary symmetrically around 50%.

## 4.2 Results for CNRM-CM6 and several exceeding thresholds

Until now, results for ToE and contributions have been presented for the probability of events exceeding the 80th percentiles of selected points belonging to $S_{90,90}^{\mathrm{CNRM\text{-}CM6}}$. In order to have a broader analysis of exceedance probabilities of compound wind and precipitation extremes, we repeat the methodology for all pairs of exceedance thresholds between the 5th and 95th percentiles (with steps of 5 percentiles) of selected points belonging to $S_{90,90}^{\mathrm{CNRM\text{-}CM6}}$. Fig. 6 displays the results obtained for the CNRM-CM6 time of emergence at the 68% confidence level, by considering marginal and dependence changes (Fig. 6a), marginal changes



only (Fig. 6b) and dependence changes only (Fig. 6c). Moreover, for each bivariate exceedance threshold, median contributions (over all sliding windows) of marginal (Fig. 6d), dependence (Fig. 6e) and interaction terms (Fig. 6f) are displayed. Results for ToE obtained at 95 % confidence level are displayed in Fig. S1 and differences of ToE are displayed in Fig. S2 of the

Supplement. When varying exceedance thresholds, different ToE results are obtained, depending on whether marginal and dependence changes are considered (Figs. 6a, b and c). ToE are found for most of the exceedance thresholds when considering both marginal and dependence changes (Fig. 6a) or marginal changes only (Fig. 6b). It is however not the case for dependence changes only (Fig. 6c), for which only specific pairs of exceedance thresholds can find times of emergence. Interestingly, these pairs correspond to very high compound wind and precipitation extremes. It indicates that dependence change plays an

important role for the probability of such high extreme events. The importance of dependence properties can also be assessed visually by comparing Figs. 6a and b. Indeed, for approximately the same pairs of exceedance thresholds as those already identified in Fig. 6c, earlier times of emergence are obtained when considering both marginal and dependence changes (Fig. 6a), than when considering only marginal changes (Fig. 6b). Concerning the median contributions over all sliding windows of the marginal (Fig. 6d), dependence (Fig. 6e) and interactions terms (Fig. 6f), results vary according to the exceedance thresholds

considered. While, for a large proportion of the exceedance thresholds, marginal properties changes contribute strongly to probability changes (Fig. 6d), dependence properties changes contribute dominantly to probability changes of very high wind and precipitation extremes (Fig. 6e). Regarding the "interaction" term, its contributions are close to 0, indicating little influence on the probability changes.

### 4.3 Results for Indiv- and Full-Ensemble version and a single exceeding threshold

We now present the results obtained for time of emergence and contributions for the Indiv- and Full-Ensemble versions for a single exceeding threshold. The methodology, previously illustrated on the CNRM-CM6 simulations, is now applied to each of the 13 models. Concerning the Indiv-Ensemble version, only one model (INMCM-5.0) had more than 5% of goodness-of-fit tests over all sliding windows rejecting the hypothesis that the copula is a good fit, and hence was excluded from the analysis (see Appendix B for further details).

We first present the results obtained for probabilities of exceeding the 80th percentiles of selected points of high values of wind and precipitation for the 1871-1900 reference period. Fig. 7 presents time series of exceedance probabilities obtained for the Indiv- and Full-Ensemble versions. Probability time series obtained for the 12 models when considering changes of marginal and dependence (Fig. 7a), marginal (Fig. 7b) and dependence properties (Fig. 7c) are displayed, as well as ToE at the 68 % confidence level for the individual models and their multi-model median estimate. Similarly, probability time series

are shown for the Full-Ensemble version in Figs. 7d, e and f. Results for time of emergence at the 95 % confidence level are presented in Fig. S3 of the Supplement. When considering future changes of both marginal and dependence properties (Fig. 7a), half of the models (6/12) detects a time of emergence at the 68% confidence level. When found, a relatively important variability of ToE across climate models is obtained (varying between 2009 (1994-2023) and 2083 (2068-2097), Fig. 7a). These different results — i.e. either a ToE is detected or not, and the important variability of the year of emergence when found — indicate

discrepancies of statistical properties of compound wind and precipitation extremes between climate models. For marginal


changes (Fig. 7b), 7 models out of 12 detect a time of emergence, within a smaller range of values. It suggests a slightly better agreement of marginal changes for future periods between models when time of emergence is defined. Moreover, models that show emergence when considering marginal changes only are not necessary those that show emergence when considering both future marginal and dependence changes. Indeed, 2 out of the 7 models emerging with marginal changes are not those from

the 6 emerging when marginal and dependence changes are taken into account (not shown). Hence, marginal changes alone are not always sufficient to make the probability signal emerge. Concerning dependence changes (Fig. 7c), 2 models out of 12 detect a time of emergence, indicating that dependence property changes for these two models influence greatly exceedance probabilities by 2100. However, it also suggests that, for most of the models, the influence of the dependence properties changes on exceedance probabilities are too small to make the probability signals go out from the reference confidence interval by 2100.

These results on the stationarity of dependence structures complement those of Vrac et al. (2022), where the ability of CMIP6 models to capture and represent significant changes in inter-variable dependencies is questioned.

Concerning the results for the Full-Ensemble version, emergence at the 68% confidence level is detected when considering marginal and dependence changes (Fig. 7d), marginal changes only (Fig. 7e) and dependence changes only (Fig. 7f) of pooled data. Emergence for the Full-Ensemble version can be partly explained by the pooling step which mechanically reduces un-

certainties in marginal and copula fitting. Then, confidence intervals, including that of the reference period, are smaller than those obtained for individual models, which leads to emergence of probability signals with small probability changes (as for probability changes induced by dependence changes only in Fig. 7f). Thus, ToE are here detected for the Full-Ensemble version despite the pooling procedure that could reduce the signal by combining models simulating different evolutions of probabilities. Results for time of emergence presented in Fig. 7 for both Indiv- and Full-Ensemble versions are summarised in Fig. S4

of the Supplement.

Now, contributions of marginal, dependence and interaction terms in probability changes are quantified for the Indiv- and Full-Ensemble versions. For the Indiv-Ensemble versions, contributions are computed for each model separately and summarised by computing the median contribution of the models. Fig. 8 displays the median contributions over all sliding windows for the 12 climate models separately, as well as for the Indiv- and Full-Ensemble versions. Time series of bivariate FAR,

relative differences and contributions along sliding windows for the Indiv- and Full-Ensemble versions are also displayed in Fig. S5 of the Supplement. Fig. 8 shows that, depending on the model, different results are obtained for the contributions to probability changes. Indeed, while some models present balanced contributions, i.e. marginal and dependence terms contributing to $\approx 50\%$ each to probability changes (e.g., CMCC-ESM2, CNRM-CM6-1 and CNRM-CM6-1-HR), other models show very unbalanced contributions, with one statistical property mainly driving the probability changes. For example, the depen-

dence term contributes dominantly ($\geq 65\%$) to probability changes for the models CanESM5, FGOALS-g3 and INM-CM-4-8, while the marginal term contributes the most for EC-Earth3, GFDL-CM4, IPSL-CM61-LR, MIROC6, MPI-ESM1-2-LR and MRI-ESM2-0. Results for Indiv- and Full-Ensemble versions are also reported, both indicating a contribution to probability changes of $\approx 60$ % from changes in marginal properties and $\approx 40$ % from changes in dependence properties. Concerning the interaction term, as obtained previously in Sect. 4.1, its contribution is close to zero for each model individually, and for Indiv-

and Full-Ensemble versions.





## 4.4 Results for Indiv- and Full-Ensemble versions and several exceeding thresholds

As previously done in Sect. 4.2, we now compute times of emergence for all combinations of exceedance thresholds between the 5th and 95th percentiles, for both Indiv- and Full-Ensemble versions, in Fig. 9. Note that, here, exceedance thresholds are now expressed in terms of percentiles to enable a comparison of results. Fig. 9a shows multi-model medians of ToE values induced by both marginal and dependence changes, i.e., results obtained for the Indiv-Ensemble version. A median value of time of emergence is obtained for any considered bivariate threshold, indicating that, for each exceedance threshold, at least one model presents an emergence. However, median ToE values show a variability depending on the bivariate exceedance thresholds. Note that, for the Indiv-Ensemble version, the number of models presenting a time of emergence can also vary from one bivariate threshold to another. For each exceedance threshold, the number of models emerging at the 68 % confidence level, as well as interquartile values, are shown in Fig. S6 of the Supplement. In particular, Fig. S6a indicates that all of the 12 models present a time of emergence for the probability of events exceeding very high precipitation and relatively low wind speed values (upper-left corner of the subplot). It suggests that all models agree on a change of the probability of occurrence of such events. This large consensus between models is not reached for events exceeding relatively low precipitation and very high wind speed values. Therefore, while all models simulate a significant increase of extreme precipitation events, it is not necessarily the case for extreme wind speed events. Results obtained for time of emergence induced by marginal properties only (Figs. 9b and S6b) are quite similar, although still indicating small differences with those obtained by considering marginal and dependence changes. Indeed, small differences of time of emergence can be observed, in particular for the upper-right area corresponding to very high wind speed and precipitation extremes. As observed in Sect. 4.1, this area corresponds to the area where dependence properties changes make emerging exceedance probability from the reference period (Fig. 9c), suggesting their importance for the probability changes of such events. This result however should not be overstated, as only ≈ 2 models show dependence changes large enough to lead to the emergence of probability (Fig. S6c).

The results of the Full-Ensemble approach are quite different from those of the Indiv-Ensemble one. For example, Fig. 9d indicates that the time of emergence for exceedance probabilities of low wind speed and high precipitation values is ≈ 2000 (while later for Indiv-Ensemble version, i.e. ≈ 2040). The results when considering marginal and dependence changes (Fig. 9d) and marginal changes only (Fig. 9e) are quite similar, indicating that changes in marginal properties mainly drive emergence of probabilities for each of the exceedance thresholds. A clear gradient of ToE values across exceedance thresholds is present: the more extreme the precipitation and the less extreme the wind speed, the sooner the time of emergence of exceedance probability. Conversely, the less extreme the precipitation and the more extreme the wind speed, the later the ToE. In fact, pooling data somehow strengthens the results for time of emergence when models agree on probability changes. Indeed, as seen previously, individual models agree in simulating a significant increase in probability of events exceeding low wind speed and high precipitation values. For ToE induced by dependence properties changes only (Fig. 9f), quite interestingly, probabilities emerge for exceedance thresholds more or less corresponding to the ones identified for Indiv-Ensemble in Fig. 9c. Although dependence properties seem to be stable over time for the majority of the models as observed in Fig. 7c, the resulting dependence structure of pooled data and its changes over sliding windows lead to obtain ToE values of exceeding probabilities.





One should also keep in mind that the reduced uncertainty for probability estimations resulting from the pooling process plays an important role in ToE detection for the Full-Ensemble version. For illustration purposes, evolutions of the bivariate distributions for the Full-Ensemble version are shown in Fig. S7 of the Supplement. Also, results for time of emergence at 95% and the number of models emerging for each exceedance threshold are displayed in Fig. S8 and Fig. S9 of the Supplement.

Median contribution of marginal, dependence and interactions terms are displayed in Fig. 10 for both Indiv- and Full-
Ensemble versions. The results obtained previously concerning the importance of the marginal properties on probability changes are here confirmed: for all exceedance thresholds, marginal properties changes contribute to more than 50 % of probability changes for both Indiv- and Full-Ensemble versions (Figs. 10a and d). Concerning dependence changes' contribution (Figs. 10b and e), the median values obtained are less than 50%, but specific pairs of exceedance thresholds highlight again the varying importance of dependence properties on exceedance probability changes: for both Indiv- and Full-versions,
median contribution of dependence properties are high for the probability changes of events exceeding high wind speed and high precipitation values. The area of exceedance thresholds for which dependence properties contribute greatly to probability changes is however greater for the Full-version (Fig. 10e) than for the Indiv-Ensemble version (Fig. 10b). Again, these results have to be directly linked with those obtained for the emergence of probabilities of such events due to dependence changes in Figs. 9c and f. Concerning the interaction term (Figs. 10c and e), contribution values are equal to 0 for both Indiv- and
Full-Ensemble versions, highlighting again the negligible role of this term in probability changes.

## 5   Results for growing-period frosts

We now apply our methodology to analyse a second type of compound events: growing-period frosts. Contrary to compound wind and precipitation extremes, for which we were interested in exceedance probabilities (i.e. both contributing variables exceeding thresholds), we are interested here in probability of growing-period frosts, i.e. the probability of having a GDD
value exceeding a threshold of 200 ($GDD \geq 200$) by the end of March — and hence characterising bud burst conditions — and having a frost in April, i.e. having $T \leq 0$. Hence, we applied our methodology described in Sect. 3 on bivariate points of GDD and minimal temperature data (one pair by year) by adapting Eq. (2) to compute the probabilities of interest. For example, for the probability of growing-period frosts in the reference period, it is computed as follows (Yue and Rasmussen, 2002):

$$p_{m,d}(0, 150) = \mathbb{P}(T \leq 0 \cap GDD \geq 200)$$
$$= F_T(0) - C(F_T(0), F_{GDD}(200)). \tag{9}$$

Although the main results are presented for a threshold of 200 °C.day, additional results for thresholds of 150 °C.day and 250 °C.day are displayed in the Supplement to assess risks of growing-period frosts for earlier and later bud burst plants.



## 5.1 Indiv- and Full-Ensemble results

We now present the results for the growing-period frosts. For the Indiv-Ensemble version, as previously, only one model (CMCC-ESM2) is excluded from the ensemble as it presents more than 5% of goodness-of-fit tests rejecting the hypothesis that fitted copulas are a good fit (see Appendix B for further details). Before computing any probability, Fig 11 displays the changes along sliding windows of the fitted bivariate distributions for the Full-Ensemble version, i.e., after pooling GDD and minimal temperature data of the different models. Clearly, a change of bivariate distributions for future periods can be visually assessed when marginal properties changes are considered (Figs. 11a and b). In particular, it presents an increase of both minimal temperature and GDD values, which could be expected in a context of global warming. The upper-left areas corresponding to probabilities of growing-period frost events ($\{G \geq 200 \cap T \leq 0\}$) are approximately similar for the first sliding windows, but their sizes increase for future periods, suggesting a greater probability of growing-period frosts induced by marginal properties changes. However, when dependence properties changes are only considered without marginal changes (Fig. 11c), bivariate distributions are quite similar and the upper-left area is almost identical in size, suggesting that the effect of dependence properties changes on growing-period frost probability is small.

Fig. 12 presents the time series of probabilities obtained for the Indiv-and Full-Ensemble versions for growing-period frost events. Results for 150°C.d and 250°C.d GDD thresholds are presented in the Supplement in Figs. S10 and S11, respectively. By considering climate models separately, a time of emergence at 68 % confidence level is detected for 11 out of 12 models when marginal properties changes are taken into account (Figs. 12a and b). Although a large majority of models agrees by simulating a significant change of growing-period frost probability with respect to the reference period, times of emergence are quite scattered, indicating differences in simulations of growing-period frosts. By considering dependence changes only (Fig. 12c), none of the 12 models within the Indiv-Ensemble presents a time of emergence, indicating that the influence of dependence changes alone is not strong enough to modify growing-period frost probabilities. For the Full-Ensemble version, changes of marginal and dependence properties (Fig. 12d) and changes of marginal properties only (Fig. 12e) lead to increase growing-period frosts probability such that time of emergence is detected at 1905 and 1906, respectively. Probability time series are quite similar, suggesting again that changes of dependence properties do not influence strongly probability of growing-period frosts. It is confirmed in Fig. 12f, for which no significant change of probability induced by dependence changes only are observed between the reference and future periods. Times of emergence obtained for growing-period frosts are summarised in Fig. S12 of the Supplement.

Fig. 13 displays the median contribution of the marginal, dependence and interaction terms to probability changes for each climate model individually and for the Indiv- and Full-Ensemble versions. For the climate models individually, as well as for the Indiv- and Full-Ensemble versions, the results are quite clear: marginal properties are the statistical properties contributing the most to probability changes of growing-period frosts. Fig. S13 shows contribution across sliding windows and hereby confirms that contributions of the dependence and interaction to change of probability are rather limited along the whole time period.



## 6   Conclusion, discussion and future work

### 6.1   Conclusions

In this study, we have presented a new methodology to assess time of emergence of compound hazards probabilities. Using a copula-based multivariate framework, we also propose to quantify the contributions of marginal and dependence properties to probability changes of hazards leading to compound events. The methodology has been applied to analyse two different climate hazards with potentially high-impacts, using a 13-member multi-model ensemble (CMIP6): compounding wind and precipitation extremes in Brittany and growing-period frosts over Central France. For each hazard, the methodology has been applied in two different versions: the Indiv-Ensemble version, for which the methodology is applied to individual climate models to derive time of emergence of probabilities and contributions of statistical properties of each model separately, and the Full-Ensemble version, for which the methodology is applied to bias-corrected and pooled data from the different models. Depending on the version, the objectives are not exactly the same: whereas the Indiv-Ensemble version enables us to estimate the uncertainty in ToE values and contributions to multivariate hazards probability changes arising from inter-model differences, the Full-version allows us to get unique ToE and contribution values accounting for the whole ensemble, that is, by taking into account the global uncertainty inherent in climate modelling.

Results for compounding wind and precipitation extremes over Brittany show that occurrence probabilities of such events are likely to increase and potentially emerge before the end of the 21st century. However, the reason of these increased probabilities can be different depending on climate models: while, for some models, probability changes are mainly driven by marginal changes only, other models give a strong importance to both marginal properties and dependence properties. It results in having a mixed importance ($\sim 65\%$ and $35\%$) of both marginal and dependence properties that contribute to probability changes within the Full-Ensemble version. These results highlight the importance of carefully taking into consideration the dependence structure when studying the evolution of probabilities of compound wind and precipitation extremes.

Concerning growing-period frosts over Central France, a large majority of models agrees on the emergence of probabilities of such events. They also agree on the dominant contribution of marginal properties changes, while the contribution of dependence properties are mostly negligible.

By analysing two different case studies, our results highlight that the importance of marginal and dependence properties to probability changes can differ from a compound hazard to another, and from one climate model to another. It thus stresses the importance of considering both marginal and dependence properties carefully, as well as their inter-model variability, to analyse the future evolution of multivariate hazards leading to compound events.

### 6.2   Discussion and perspectives

In this study, emergence of probabilities of multivariate hazards has been investigated with respect to the baseline period 1871-1900. This period can be considered as representative of the beginning of the industrial era (e.g., Hawkins et al., 2020) and can hence be of interest to assess if anthropogenic climate change has contributed to an emergence of probability of multivariate hazards. However, other baseline periods could have been chosen, such as more recent ones which would provide useful





results for adaptation planning (e.g., Ossó et al., 2022). Of course, depending on the chosen baseline period, the estimated natural variability that serves as reference for assessing changes would be different, and thus would affect the ToE results.

As an illustration, Fig. S14 shows results from a quick sensitivity experiment for the time of emergence of probabilities of compounding wind and precipitation depending on the choice of the baseline period for the CNRM-CM6 model. It illustrates that results of emergence can vary strongly depending on the chosen baseline period. In addition to modifying the potential time of emergence, the choice of the baseline period can also influence the results of contributions from the statistical properties changes (not shown), as these statistical changes are also assessed with respect to the baseline period.

Moreover, in this study, time of emergence of probability signals is defined as the year or time period for which the probability signal *permanently* excesses a certain threshold (e.g., Hawkins and Sutton, 2012; Maraun, 2013; Hawkins et al., 2020). As the Earth's climate system is highly nonlinear and non-monotonic, detecting the emergence of a signal in this way can be limited depending on the climate signal under study. Analysing "periods of emergence" (PoE) instead of time of emergence may be more relevant to rather describe specific periods where probability signals emerge significantly — but temporarily —

from reference natural variability. This notion of PoE would better highlight not only the non-linearities of the CE changes but also the differences of evolution of probability between climate models, as it was observed for growing-period frosts in Sec. 5. Indeed, in Fig. 12, while some climate models reach their highest growing-period frosts probability for the late 21st century, other climate models present a decrease of probability to 0 for the end of the century after having reached maximum growing-period frosts probability earlier. In other words, probabilities for future periods may differ, not permanently,

but only temporarily from the estimated probability associated with natural variability. This could justify the development, the investigation and the use of the notion of temporary periods of emergence.

In addition, changes in marginal properties of the different variables and their contributions to probability changes have been assessed together, i.e., without separating the changes and contributions from wind and precipitation, nor those from GDD and minimum temperature. Thus, it does not allow us quantifying by how much individual variables' changes drive

probability changes. Some studies already concluded about the importance of individual variables in the change of occurrence of multivariate hazards (e.g., Manning et al., 2018; Brunner et al., 2021; Calafat et al., 2022). Our methodology can however be easily adapted to quantify such information by keeping fixed marginal properties of only one contributing variable and assess probability changes. By doing this for the different variables in turn, the contribution of marginal changes to probability changes would be decomposed according to individual variables changes.

This study shows that both univariate and multivariate properties can be essential in determining CE properties. However, despite substantial improvements in climate modelling, climate simulations often remain biased compared to observations or reanalyses in terms of both univariate and multivariate properties (e.g., Cannon, 2018; Vrac, 2018; François et al., 2020). This could have major consequences on the ability of climate models to simulate compound events accurately (Zscheischler et al., 2019; Villalobos-Herrera et al., 2021; Vrac et al., 2021; Ridder et al., 2021), and then on the resulting analyses involved in

decision-making processes. A few multivariate bias correction methods, i.e. statistical methods that are able to adjust both univariate and multivariate properties of simulations with respect to reference dataset, have been recently developed (e.g., Cannon, 2018; Guo et al., 2019; Mehrotra and Sharma, 2019; Robin et al., 2019; Vrac and Thao, 2020; François et al., 2021).


However, such MBC methods are designed to adjust the whole statistical distribution of climate simulations, and their abilities to increase the realism of specific parts of the statistical distribution (such as multivariate extremes) have never been tested, while it can be crucial for specific CEs. This is therefore an important perspective and the methodology developed in the present study could be a way to evaluate the consequences of MBC methods, e.g., in terms of ToE and contributions of marginal and dependence properties.

It has to be noted that uncertainty in probabilities of multivariate hazards has been assessed by considering uncertainty in both statistical fitting procedures and model-to-model differences. However, uncertainty arisen from internal climate variability, i.e., from the inherent chaotic nature of the climate system, has not been investigated. Assessing and analysing these uncertainties is however key to better characterise them and thus provide useful information for policy-makers (Raymond et al., 2022; Bevacqua et al., 2022). Future extensions of the framework presented herein could thus focus on using multimodel large-ensemble simulations to assess more robustly probabilities of hazards, contributions of statistical properties changes to their emergence, and their associated uncertainties resulting from both internal variability and structural model differences.

It is also important noting that the role of physical drivers of multivariate hazards has not been investigated in this study. Indeed, recent studies highlight the importance of large-scale climate modes (e.g., De Luca et al., 2020b; Singh et al., 2021b) and atmospheric circulation regimes (e.g., Faranda et al., 2020; Jézéquel et al., 2020; Vrac et al., 2021) on compound and extreme events. Understanding the influences of physical drivers and their changes on the statistical features and probabilities of multivariate hazards is a key research which has important implications for predicting their occurrence and characterising their impacts.

As mentioned in Sect. 1, the present methodology has been developed and applied in a ToE framework that is different from attribution. We have not considered factual and counterfactual worlds with different forcings to assess the effects of climate change on multivariate hazards probabilities. Adapting and applying our methodology in an attribution setting is thus an interesting perspective that would complement the existing multivariate event attribution framework recently developed (e.g., Kiriliouk and Naveau, 2020; Zscheischler and Lehner, 2021). In addition to attributing changes of compound events, our methodology would permit to quantify the underlying contributions of the changes in marginal and dependence properties, hence better characterising the statistical features of climate change.

*Code availability.* Custom codes developed for the analyses are publicly available at https://github.com/bastien-francois/ToE_CE.

*Data availability.* CMIP6 climate model data can be downloaded through the Earth System Grid Federation portals. Instructions to access the data are available here: https://pcmdi.llnl.gov/CMIP6/Guide/dataUsers.html, last access: 23 January 2022.



## Appendix A: Procedure for confidence intervals estimation

Confidence intervals of bivariate exceedance probabilities are estimated by combining the confidence intervals from the fitted
parameters for both marginal distributions and copulas. For both marginal distributions and copula, the fitted parameters and
their 68 % (resp. 95 %) confidence intervals are estimated using MLE (as described in Appendix B) and profile likelihood
(e.g., Venzon and Moolgavkar, 1988; Hofert et al., 2012). Estimating the 68 % (resp. 95 %) confidence intervals for bivariate
exceedance probabilities consists in: (i) resampling uniformly and independently the fitted parameters of the two marginal
distributions within their 68 % (resp. 95 %) profile likelihood confidence intervals, (ii) computing the bivariate exceedance
probability using the resampled parameters for marginal distributions and the copula parameter estimated using MLE, (iii)
repeating the two previous step 100 times to construct a sampling distribution for the bivariate exceedance probability, (iv)
searching which combinations of the resampled parameters lead to the 16 % and 84 % (resp. 2.5 % and 97.5 %) percentiles of
the re-estimated bivariate exceedance probabilities, (v) using the copula parameter uncertainty, estimating the 68 % (resp. 95
%) confidence intervals of the 16 % and 84 % (resp. 2.5 % and 97.5 %) percentiles of the bivariate exceedance probabilities.
The lower and upper bounds of these two confidence intervals define the final confidence interval combining both marginal
and copula parameters uncertainty.

## Appendix B: Marginal and copula fitting

For the fitting of the marginal distributions, we considered the Akaike information criterion (AIC) to select the best families
among Gaussian, generalized extreme value and generalized Pareto distributions. The marginal distributions of wind speed
and precipitation beyond the selection thresholds were modeled by generalized Pareto distributions. For growing-period frost
events, the marginal distributions of the GDD indices were modeled using Gaussian distributions. We modeled the negative of
the minimal temperatures using GEV distributions and transformed back.

For fitting of the copulas, marginal distributions are transformed into uniform distribution using normalized ranks (e.g.,
Salvadori et al., 2011; Serinaldi, 2015; Bevacqua et al., 2019). This procedure is common for copula analysis as it allows
to perform appropriate goodness of fit tests (Genest et al., 2009). In this study, four Archimedean copulas (Clayton, Frank,
Gumbel and Joe) are considered. These copulas have been widely used in hydrology and climate studies (e.g., Zscheischler
and Seneviratne, 2017; Liu et al., 2018b; Tavakol et al., 2020) and allow the dependence structure to be modelled with a single
parameter that determines the strength of the dependence. Moreover, the four Archimedean copulas differ in how they model
dependence structures. For instance, the Gumbel and Joe copulas have upper tail dependence, which means that they are able
to model correlated extremes. The Clayton copula has lower tail dependence and the Frank copula has no tail dependence.
A complete overview of copula families, their related functions and the range of their parameters is offered by Sadegh et al.
(2017). For each climate model and each sliding window, the best copula family is determined using the Akaike Information


Criterion. Copulas were fitted through maximum likelihood estimators (MLE) using the copula (Hofert et al., 2020) and VineCopula: (Schepsmeier et al., 2016) R-packages. Goodness of fit are tested based on the White's information matrix equality (White, 1982; Huang and Prokhorov, 2014) implemented in the R package VineCopula (Schepsmeier et al., 2016). To evaluate exceedance probabilities, we select the copula family that has been the most selected along all the sliding windows and for which less than 5% of the goodness of fit tests conclude to the rejection that data fits well the considered copula distribution. For the Indiv-Ensemble version, climate models for which more than 5% of the goodness of fit tests conclude to a rejection are excluded.

## Appendix C: Transformation of wind and precipitation data using CDF-t

As selection thresholds for wind and precipitation extremes are not the same for all the climate models, we need to transform selected wind and precipitation data. For each model, bivariate points of high values are selected using the individual 90th percentiles of wind and precipitation variables. Then, the selected bivariate data from the different models are adjusted with respect to a model taken as reference, using a univariate bias correction technique called the "Cumulative Distribution Function – transform" method (CDF-t, Michelangeli et al., 2009; Vrac et al., 2012). The CDF-t method allows to correct the univariate distribution of a modeled climate variable via a quantile-quantile method that takes into account potential changes of the univariate distribution in the correction procedure. By choosing a model as reference (CNRM-CM6), we use here the CDF-t method to transform marginal properties of selected wind and precipitation values of each climate dataset with respect to CNRM-CM6. This way, marginal distributions of wind and precipitation extremes are similar between the different climate models and are thus more consistent with each other. We consider the 1871-1900 sliding window as reference period for the calibration of the bias correction. Once data have been transformed for each climate model, bivariate wind and precipitation extreme values from the different models can be pooled and the Full-Ensemble methodology can be applied.

*Author contributions.* MV had the initial idea of the study. MV and BF designed the experiments and protocols. BF made all computations and figures. BF and MV made the analyses and interpretations. BF wrote the first complete draft of the article, with inputs, corrections and additional writing contributions from MV.

*Competing interests.* The authors declare that they have no competing interests.

*Acknowledgements.* We acknowledge the World Climate Research Programme's Working Group on Coupled Modelling, which is responsible for CMIP, and we thank the climate modeling groups (listed in Table 1 of this paper) for producing and making available their models outputs. For CMIP, the U.S. Department of Energy's Program for Climate Model Diagnosis and Intercomparison provides coordinating support and led development of software infrastructure in partnership with the Global Organization for Earth System Science Portals. The authors





acknowledge support from the EUR IPSL Climate Graduate School project managed by the ANR under the "Investissements d'avenir" programme with the reference ANR-11-IDEX-0004-17-EURE-0006, the European Union's Horizon 2020 research and innovation programme via the "XAIDA" project (Grant agreement No. 101003469), as well as from the "COESION" project funded by the French National program LEFE (Les Enveloppes Fluides et l'Environnement).



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

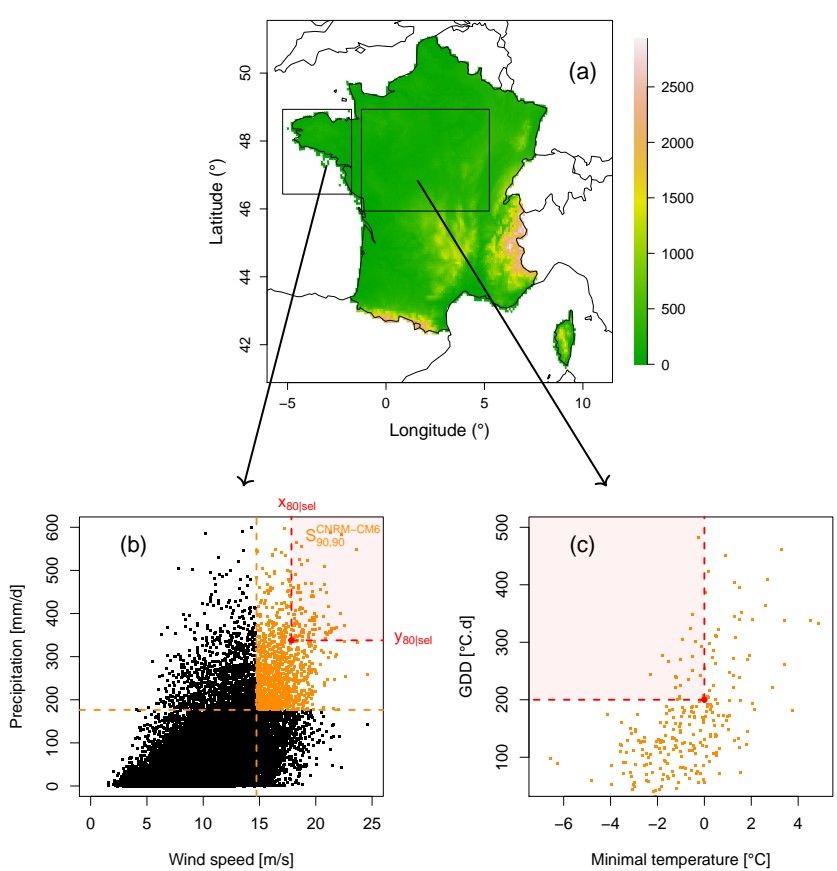

**Figure 1.** (a) Map of France with the regions of interest in boxes. Scatterplots of CNRM-CM6 (b) DJF compounding wind and precipitation in Brittany and (c) miniminal temperature in April and GDD values by the end of March over Central France for the 1871-2100 period. Parametric fitting for marginal and dependence over the 30-years sliding windows spanning the 1871-2100 period are performed to bivariate points in orange. For compounding wind and precipitation, these points correspond to high values of wind and precipitation data belonging to $S_{90,90}^{\text{CNRM-CM6}}$, i.e. simultaneously exceeding the individual 90th percentiles of the 1871-1900 reference period. Bivariate exceedance probabilities are then computed for varying exceedance thresholds between the 5th and 95th percentile of wind speed and precipitation already belonging to $S_{90,90}^{\text{CNRM-CM6}}$ (for more details, see Sect. 4). The red area contains bivariate points exceeding the 80th percentiles of points already belonging to $S_{90,90}^{\text{CNRM-CM6}}$. For growing-period frosts, exceedance thresholds of interest for minimal temperature and GDD index are fixed to values of 0°C and 200°C.day, respectively.

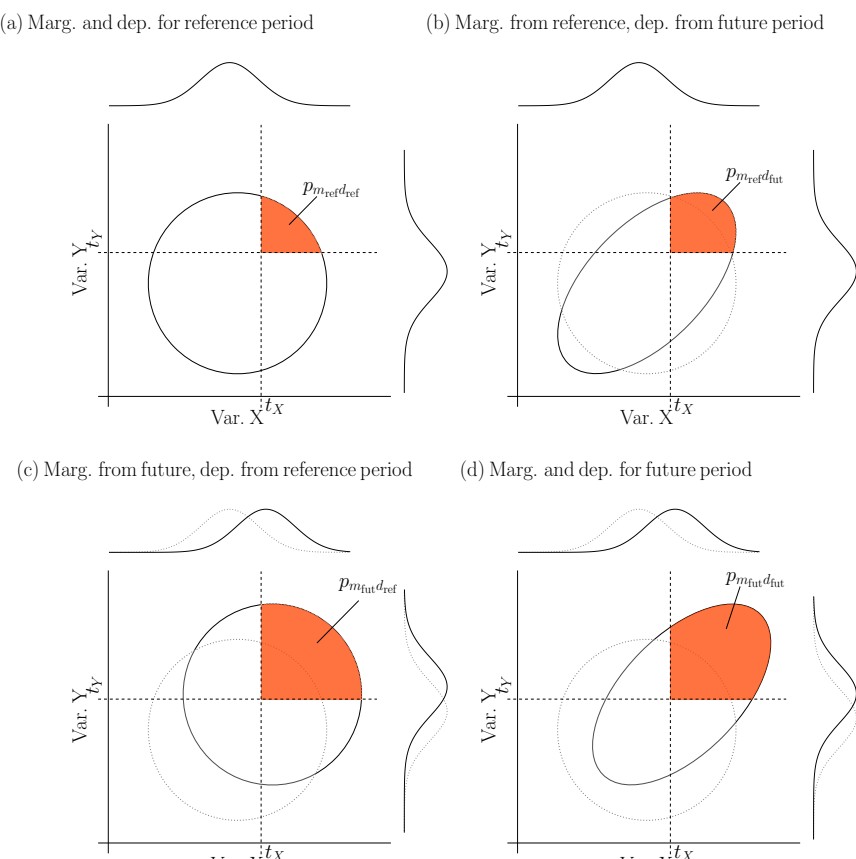

**Figure 2.** Illustration of the influence of marginal and dependence properties on bivariate exceedance probabilities for an artifical distribution of two contributing variables $X$ and $Y$ during (a) the reference period and (d) a future period with a shift in means and an increase in dependence between the variables. The distribution of the two contributing variables (b) with marginal properties from the reference period and dependence structure from the future period, and (c) with marginal properties from the future period and dependence structure from the reference period. Orange areas show bivariate exceedance probabilities for the thresholds $(t_X, t_Y)$ of the two contributing variables.





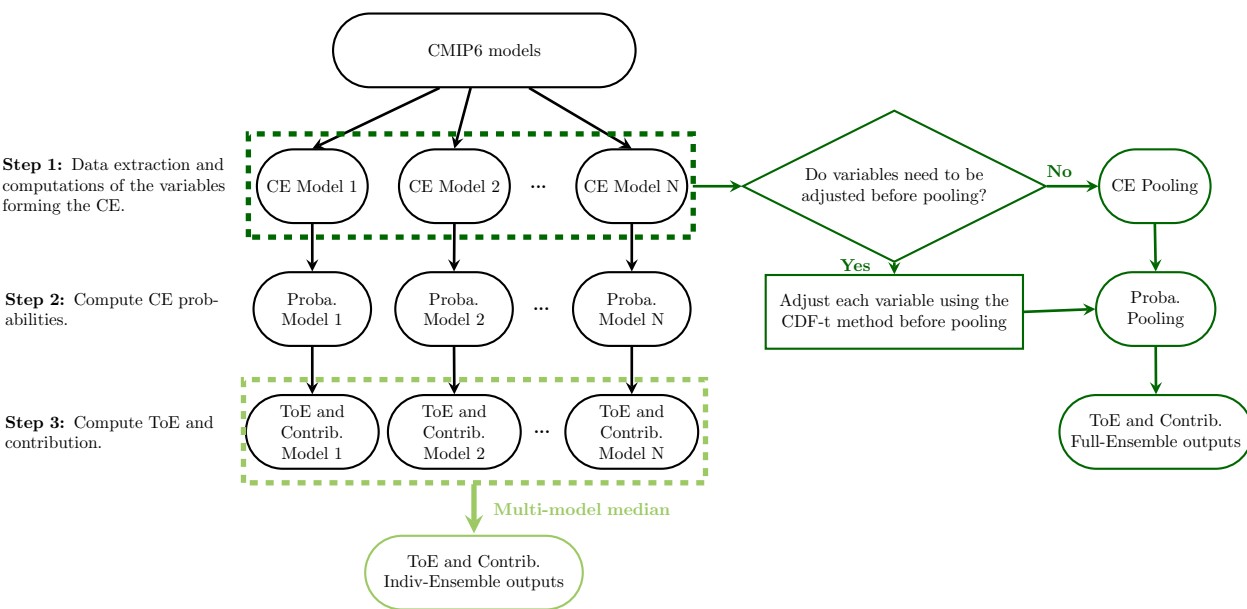

**Figure 3.** Flowchart for the computations of time of emergence and contributions for Indiv- and Full-Ensemble versions.


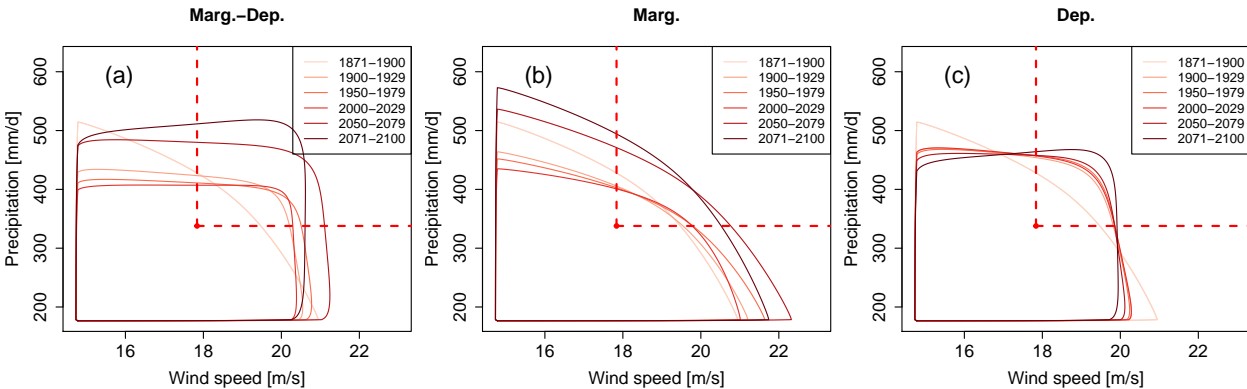

**Figure 4.** Change of winter (December-to-February) bivariate wind and precipitation extremes distributions in Brittany based on CNRM-CM6 simulations due to (a) future marginal and dependence changes ("Marg.-dep."), (b) future marginal changes while keeping dependence of the reference period ("Marg.") and (c) dependence changes while keeping marginal of the reference period ("Dep."). For the bivariate distributions, contour lines encompassing 90 % of all data points are shown. A selection of six 30-years sliding windows is presented using a color gradient from light (1871-1900) to dark (2071-2100). The red dashed lines characterises the bivariate exceeding thresholds defined here as the 80th quantile of each variable.

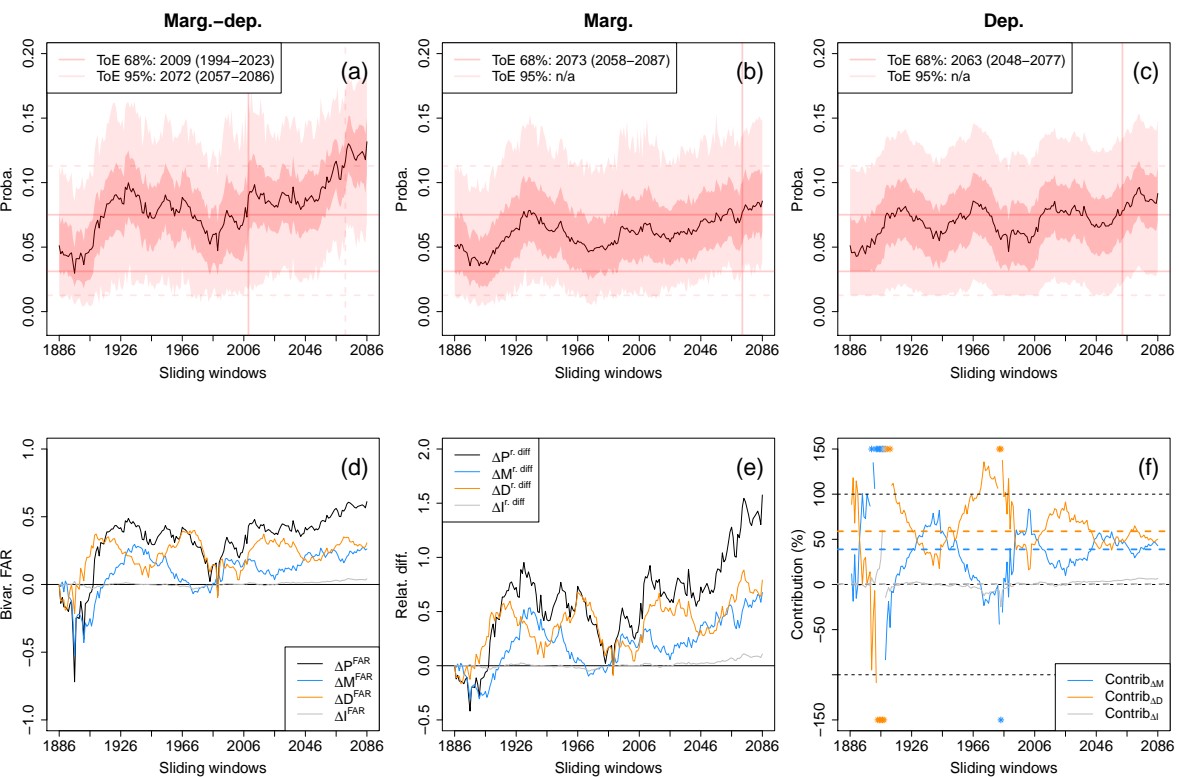

**Figure 5.** (a-c) Probability changes and time of emergence of compound wind and precipitation extremes ($\mathbb{P}(X > x_{80|\text{sel}} \cap Y > y_{80|\text{sel}} \mid (X,Y) \in S_{90,90}^{\text{CNRM-CM6}})$) based on CNRM-CM6 simulations due to changes of (a) both marginal and dependence properties, (b) marginal properties only, and (c) dependence properties only. The shaded bands indicate 68% and 95% confidence intervals of the probabilities. Evolutions of (d) the bivariate fraction of attributable risk (FAR), (e) relative difference of probabilities with respect to the reference period (1871-1900) and (f) contribution of the marginal, dependence and interaction terms to probability values. Median contributions computed over all sliding windows are displayed with dashed lines. Asterisks indicate values lying outside the plotted range. Not-applicable (n/a) is indicated when no time of emergence is detected.



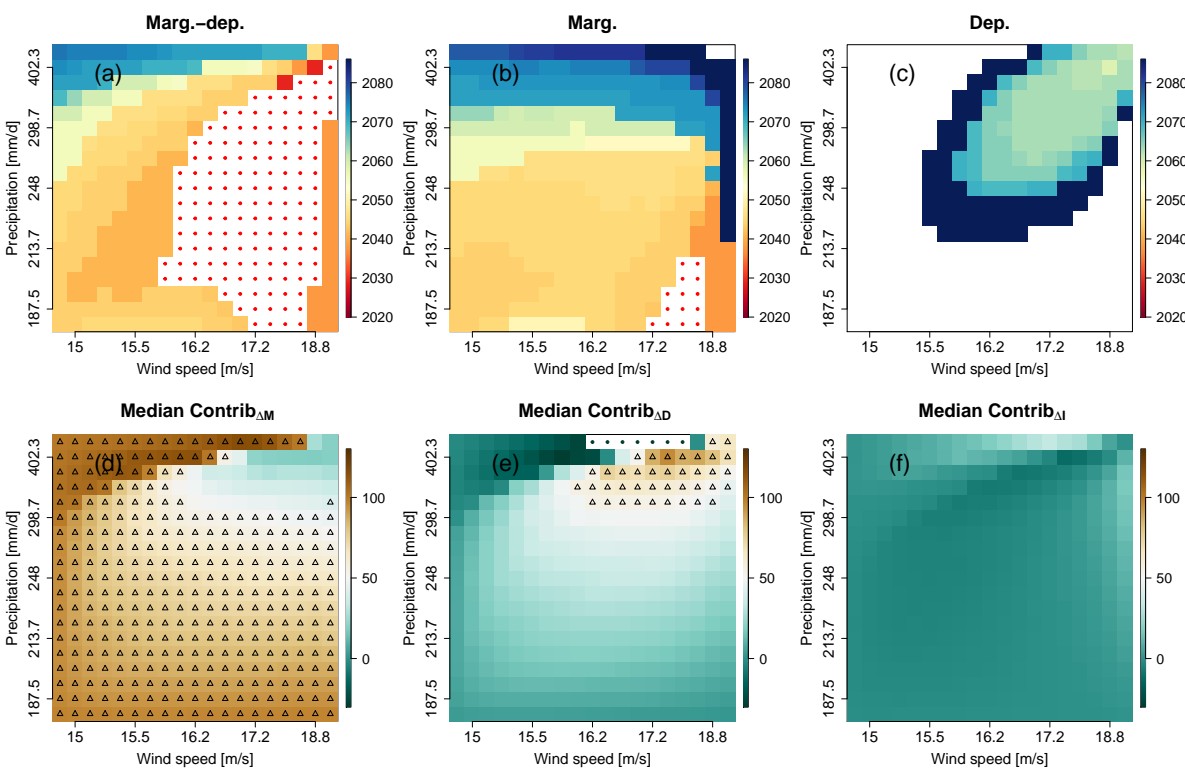

**Figure 6.** CNRM-CM6 (a-c) time of emergence (at 68% confidence level) for compound wind and precipitation extremes due to changes of (a) both marginal and dependence properties, (b) marginal properties only, and (c) dependence properties only. White cells indicate that no time of emergence is detected, while white cells with red points indicate ToE values before 2020. (d-f) Matrices of median contributions of the (d) marginal, (e) dependence and (f) interaction terms. Results are presented for varying exceedance thresholds between the 5th and 95th percentile of compound wind and precipitation extremes data. Upper triangles show where contribution $\geq 50$ %.


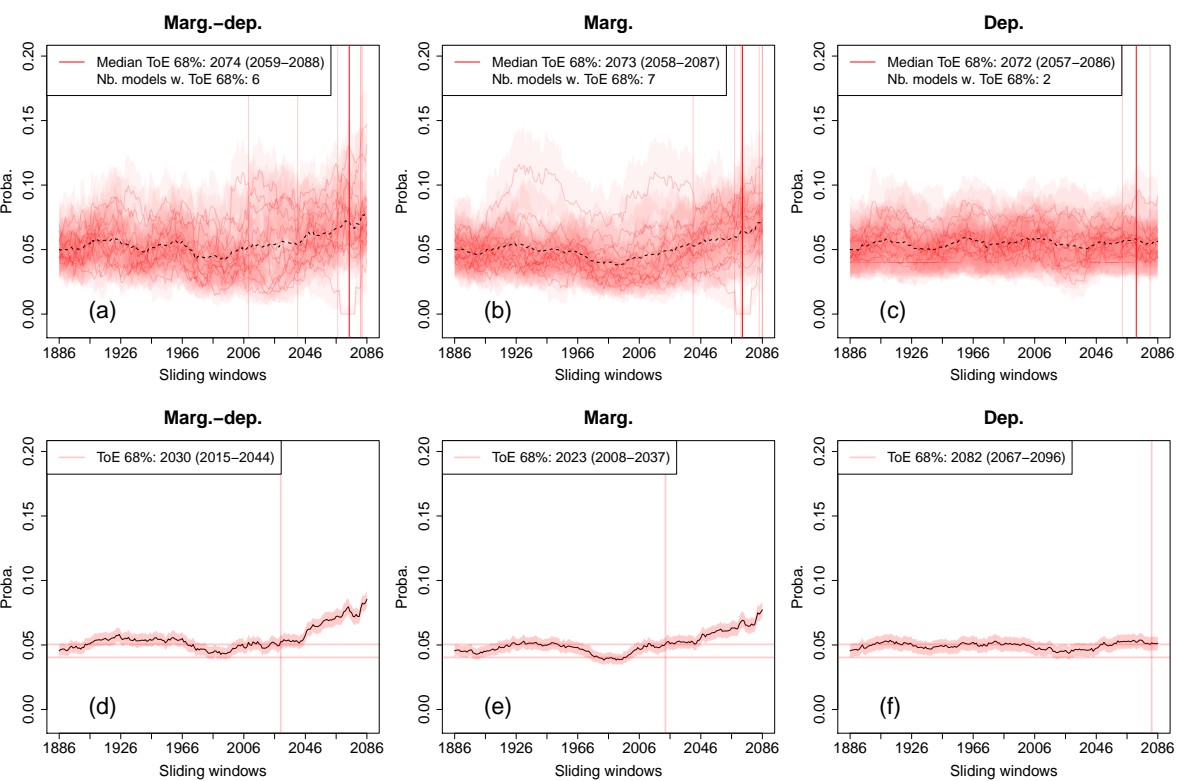

**Figure 7.** Probability changes and time of emergence (at 68%) of compound wind and precipitation extremes (exceeding the individual 80th percentiles of selected points of high values) for (a-c) Indiv- and (d-f) Full-Ensemble versions due to changes of (a,d) both marginal and dependence properties, (b,e) marginal properties only, and (c,f) dependence properties only. The shaded bands indicate 68% confidence intervals of the probabilities. For (a-c), individual time of emergence for the different models within the ensemble are displayed when defined (vertical light red lines), as well as the corresponding median time of emergence (vertical red line). For information purpose, multi-model mean exceedance probability time series are also plotted (black dotted lines).


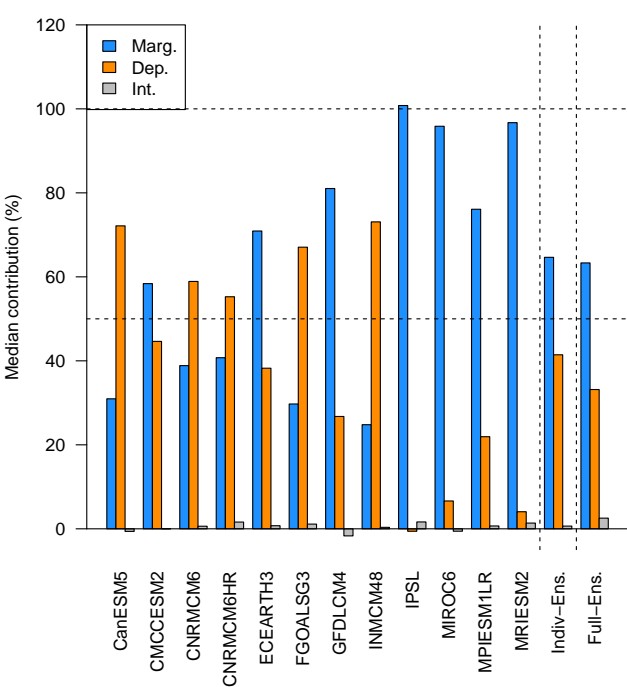

**Figure 8.** Median contribution, over all sliding windows, of the marginal, dependence and interaction terms to overall probability changes for the 12 individual CMIP6 models, and for Indiv- and Full-Ensemble versions.

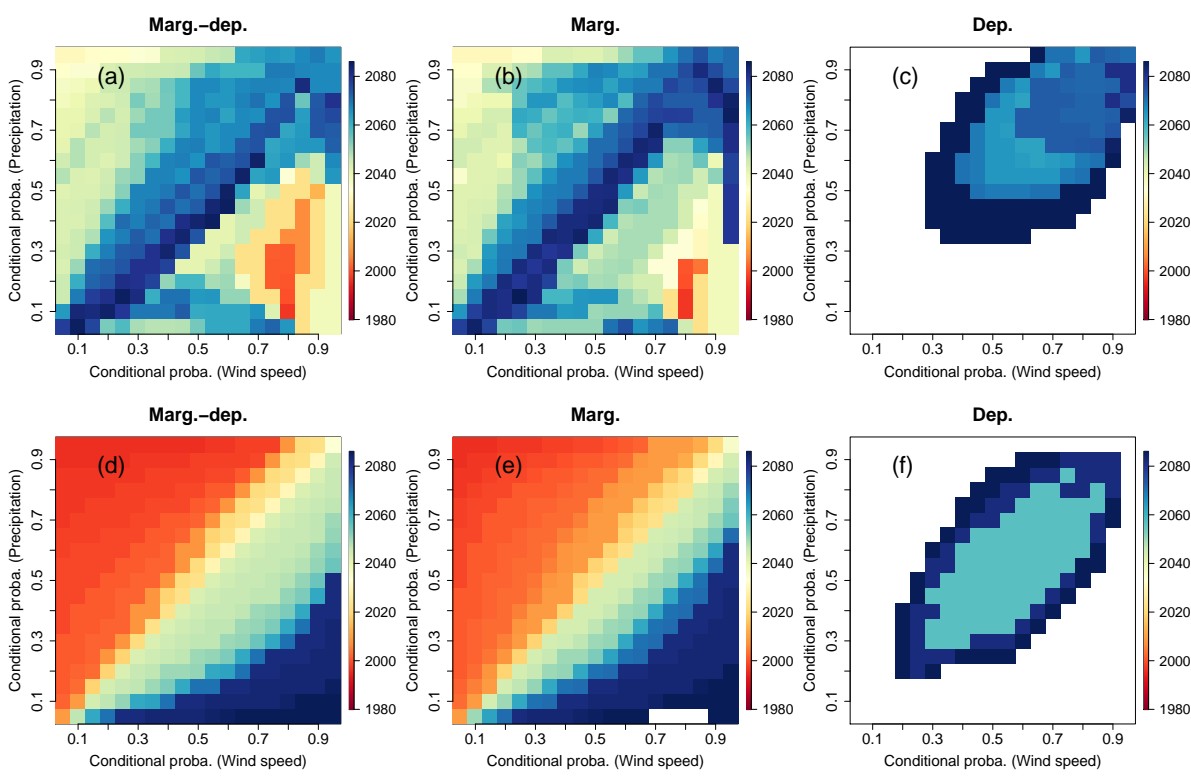

**Figure 9.** Time of Emergence (at 68% confidence level) matrices of compound wind and precipitation extremes due to changes of (a, d) both marginal and dependence properties, (b, e) marginal properties only, and (c, f) dependence properties only. Results are displayed for (a-c) the Indiv- and (d-f) Full-Ensemble versions for varying exceedance thresholds between the 5th and 95th percentile of compound wind and precipitation extremes data. For each subplot, white indicates that no time of emergence is detected.

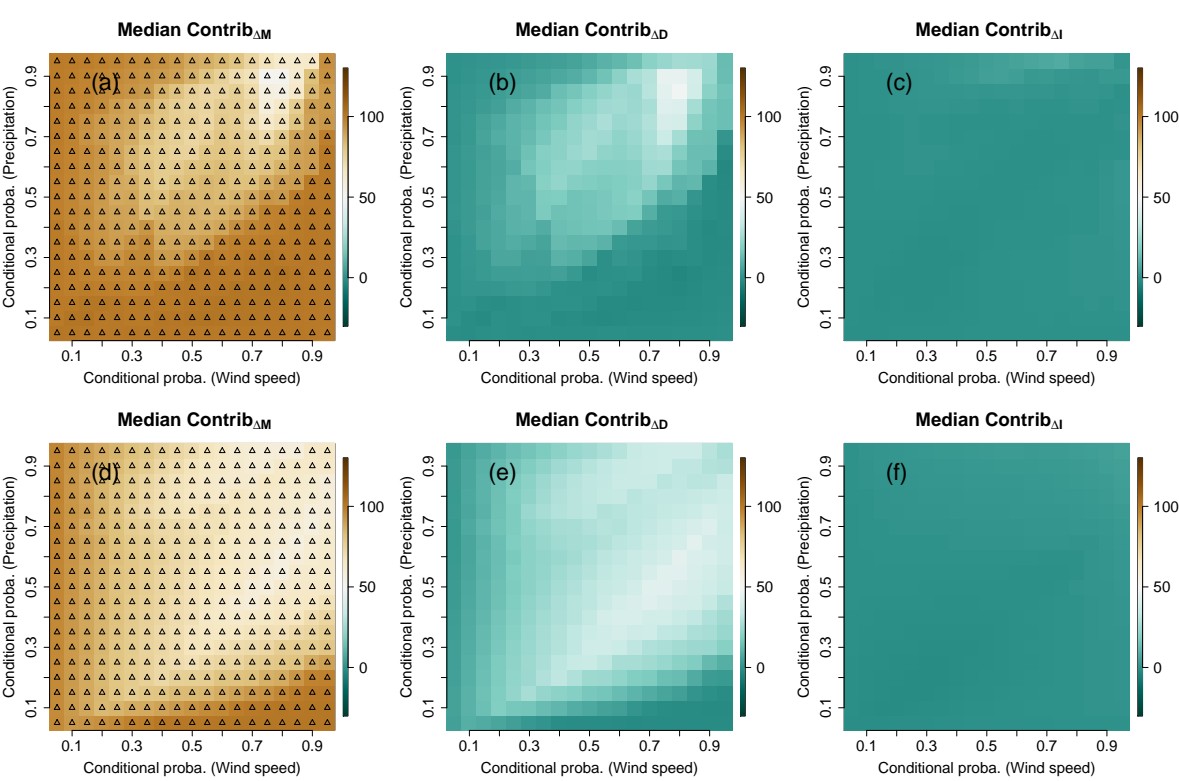

**Figure 10.** Median contributions of (a, d) marginal, (b, e) dependence and (c, f) interaction terms for (a-c) Indiv- and (d-f) Full-Ensemble versions. Results are presented for compound wind and precipitation extremes with varying exceedance thresholds between the 5th and 95th percentile. Upper triangles show where contribution $\geq$ 50 %.


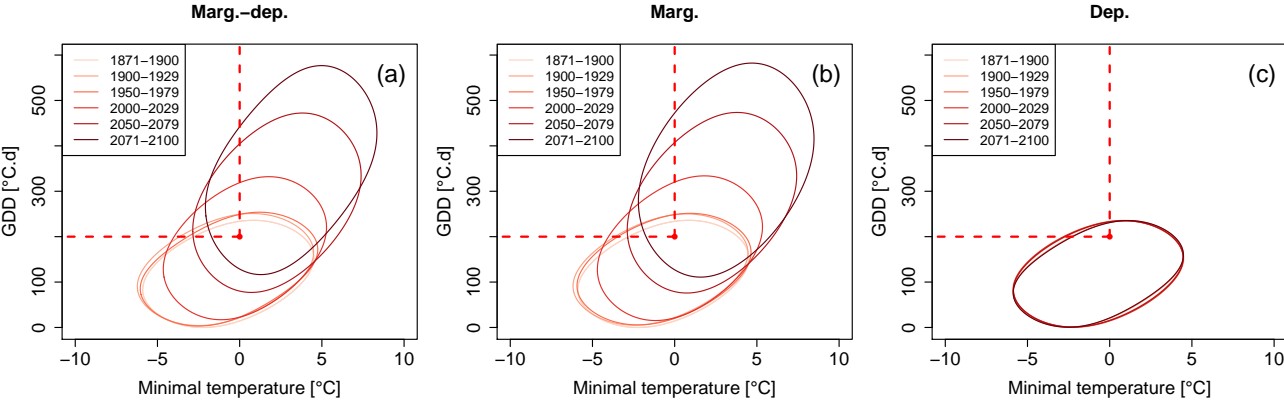

**Figure 11.** Changes of minimal temperature vs. GDD distributions in Central France for the Full-Ensemble version due to (a) marginal and dependence changes ("Marg.-dep."), (b) marginal changes while keeping dependence of the reference period ("Marg.") and (c) dependence changes while keeping marginal of the reference period ("Dep."). For the bivariate distributions, contour lines encompassing 90 % of all data points are shown. A selection of six 30-years sliding windows is presented using a color gradient from light (1871-1900) to dark (2071-2100).


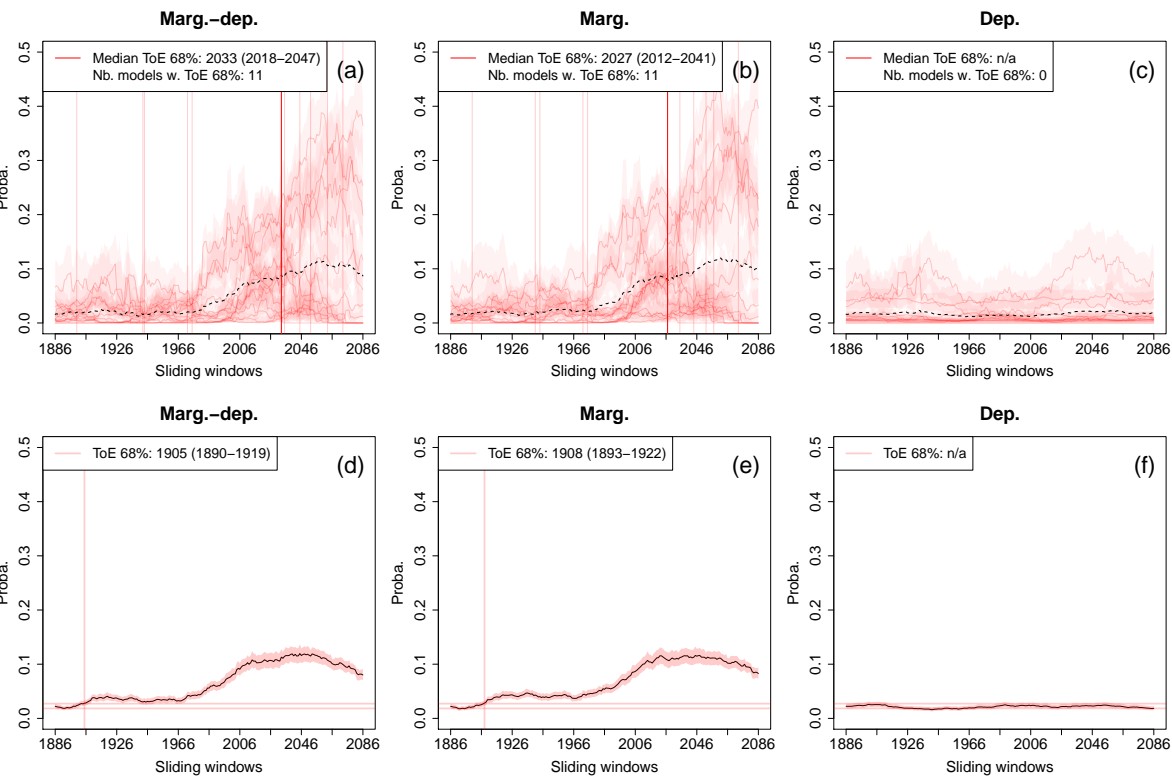

**Figure 12.** Probability changes and times of emergence (at 68%) of growing-period frosts (GDD $\geq$ 200 °C.d and minimal temperatures $\leq$ 0 °C) for (a-c) Indiv- and (d-f) Full-Ensemble versions due to changes of (a, d) both marginal and dependence properties, (b, e) marginal properties only, and (c, f) dependence properties only. The shaded envelops indicate 68% confidence intervals of the probabilities. For (a-c), individual time of emergence for the different models within the ensemble are displayed when defined (vertical light red lines), as well as the corresponding median time of emergence (vertical red line). For information purpose, multi-model mean exceedance probability time series are also plotted (black dashed lines). Not-applicable (n/a) is indicated when no time of emergence is detected.

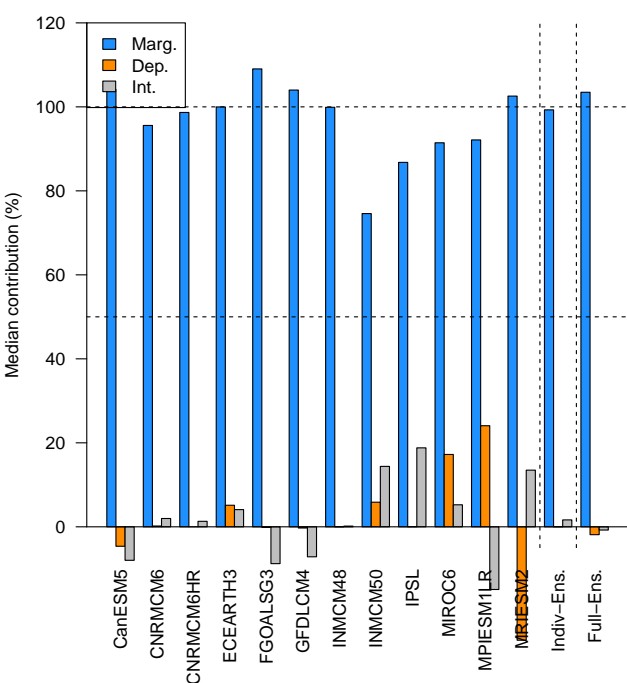

**Figure 13.** Median contribution, over all sliding windows, of the marginal, dependence and interaction terms to overall probability changes for the 12 individual CMIP6 models, and for Indiv- and Full-Ensemble versions.



**Table 1.** List of CMIP6 simulations used in this study, their run, approximate horizontal resolution and references.

| Model | Institution | Spatial res. (lon. × lat.) | Data reference |
|---|---|---|---|
| CanESM5 | Canadian Centre for Climate Modelling and Analysis, Canada | $2.81° \times 2.81°$ | Swart et al. (2019) |
| FGOALS-g3 | Chinese Academy of Sciences, China | $2.00° \times 2.25°$ | Li (2019) |
| CNRM-CM6-1 | Centre National de Recherches Meteorologiques, Meteo-France, France | $1.41° \times 1.41°$ | Voldoire (2019) |
| CNRM-CM6-1-HR | Centre National de Recherches Meteorologiques, Meteo-France, France | $0.50° \times 0.50°$ | Voldoire (2018) |
| GFDL-CM4 | Geophysical Fluid Dynamics Laboratory, USA | $1.25° \times 1°$ | Guo et al. (2018) |
| INM-CM4-8 | Institute for Numerical Mathematics, Russia | $2° \times 1.5°$ | Volodin et al. (2019a) |
| INM-CM5-0 | Institute for Numerical Mathematics, Russia | $2° \times 1.5°$ | Volodin et al. (2019b) |
| IPSL-CM6A-LR | Institut Pierre-Simon Laplace, France | $2.50° \times 1.26°$ | Boucher et al. (2018) |
| MIROC6 | JAMSTEC, AORI, NIES, R-CCS, Japan | $1.41° \times 1.41°$ | Shiogama et al. (2019) |
| MPI-ESM1-2-LR | Max Planck Institute for Meteorology, Germany | $1.88° \times 1.88°$ | Wieners et al. (2019) |
| MRI-ESM2-0 | Meteorological Research Institute, Japan | $1.13° \times 1.13°$ | Yukimoto et al. (2019) |
| CMCC-ESM2 | Centro Euro-Mediterraneo per i Cambiamenti, Italy | $1.25° \times 0.94°$ | Cherchi et al. (2019) |
| EC-Earth3 | EC-Earth-Consortium | $0.70° \times 0.70°$ | EC-Earth (2019) |