# Peer review of "Time of Emergence of compound events: contribution of univariate and dependence properties"

_Natural Hazards and Earth System Sciences, 2022_

## Author Comment (AC1)

**_Response to Referee Comment 1 on_ "Time of Emergence of compound events: contribution of univariate and dependence properties" _by_ Bastien François et al.**

**_Jakob Zscheischler_**

**General comments:**

**Comment:**

François et al present a very thorough analysis of the time of emergence of compound events. The paper introduces the concept and illustrates it for two types of compound events, compound wind and precipitation extremes and false spring events. Overall this is a timely and useful study and proposes convincing ideas on how to disentangle the contribution of marginals and dependence in trends of compound event occurrence. I find it particularly interesting that in the first example changes in the dependence matter (for some models) for ToE, whereas in the second model dependence changes are irrelevant/do not occur.

**Response:**

We would like to thank Jakob Zscheischler for his very positive comments and the detailed questions. All the comments and our point-by-point responses are given below.

**Comment:**

While the paper is very thorough, it is also somewhat lengthy, so the authors might want to consider shortening some aspects to improve readability.

**Response:**

We agree with this comment. As suggested by Anonymous Referee #2 in the comment 2), we are relocating the application of our methodology in its "Full-version" to the supplementary material. This implies that the explanations, results, figures and appendices related to the Full-version are removed from the main body of the article, thus shortening the article and improving readability, as desired. All the modifications will be detailed in the response to the second comment of Anonymous Referee #2.

**Comment:**

I have a few (mostly minor) comments that should be considered before publication.

Line 65: "Recently, Abatzoglou et al. (2020) even showed, using reanalysis data, that changes in dependence properties have been more important than changes in univariate properties in the recent decades." Not sure that was really shown in that study.

**Response:**

We agree and propose to be more explicit in the following correction (in red and blue):

> "*Recently, Abatzoglou et al. (2020) even showed*  ***that, in the recent decades, changes in multivariate annual climatic conditions (water deficit, evapotranspiration, minimum and maximum temperature) with respect to a reference climate state have been more important than changes in univariate annual climatic conditions for a large portion of the Earth.***"

**Comment:**

Fig. 5: it looks like the chosen window length is a bit too small to obtain robust results (there is very high variability in the time series, leading to large uncertainties regarding the ToE). 30 years is very limited for studying compound events. I would be interested to know whether you get smoother curves if you increase the window length and thus sample size.

**Response:**

As mentioned by the referee, our methodology can be applied by considering a larger window length than 30 years.

New Fig. S15 (below) shows the results we obtain by analyzing probability changes for the CNRM-CM6 simulations with:

- a 30-year sliding window (S15a-c, same results as those presented in Fig. 5a, b and c, baseline period: 1871-1900)

- a 40-year sliding window (baseline period: 1871-1910, S15d-f)

- a 50-year sliding window (baseline period: 1871-1920, S15g-i)

- a 60-year sliding window (baseline period: 1871-1930, S15j-l).

[Figure]

New Figure S15: Probability time series and time of emergence of compound wind and precipitation extremes ($P(X > x_{80|sel} \cap Y > y_{80|sel} \mid$ CNRM-CM6 $(X, Y) \in S^{\text{CNRM-CM6}}_{90,90})$ based on CNRM-CM6 simulations due to changes of (a, d, g, j) both marginal and dependence properties, (b, e, h, k) marginal properties only, and (c, f, i, l) dependence properties only. Results are displayed for probabilities computed by using (a-c) 30-year, (d-f) 40-year, (g-i) 50-year and (j-l) 60-year windows sliding over the period 1871-2100. In each panel, the first sliding window is considered as the baseline period. The shaded bands indicate 68% and 95% confidence intervals of the probabilities. Not-applicable (n/a) is indicated when no time of emergence is detected.

Indeed, by increasing the window length, smoother curves of probability time series are obtained, whether marginal and/or dependence changes are considered (compare Figs. S15a-c with Figs. S15d-f, g-i and j-l). ToE results are changed and will be discussed in the response to the next comment. Also, increasing the window length results in obtaining probability curves that are flatter, especially when considering the changes of dependence only (Figs. S15c, f, i and l). This can be explained by the fact that, by increasing the size of the sliding windows, more years of data are analyzed together. In a transient climate context, this results in mixing different climate conditions and, thus, different statistical properties. Changes of statistical properties between the baseline period and the sliding windows are then attenuated, which is particularly true for the dependence structure for which the climate change signal is less pronounced than for the marginal properties.

We also derived new Fig. S16 that shows the evolutions of bivariate FAR, relative differences and contribution for the CNRM-CM6 simulations with the different window lengths.

[Figure]

New Figure S16: Evolutions of (a, d, g, j) the bivariate fraction of attributable risk (FAR), (b, e, h, k) relative difference of probabilities with respect to the baseline periods and (c, f, i, l) contribution of the marginal, dependence and interaction terms to probability values. Median contributions computed over all sliding windows are displayed with dashed lines. Results are displayed for probabilities computed by using (a-c) 30-year, (d-f) 40-year, (g-i) 50-year and (j-l) 60-year windows sliding over the period 1871-2100. In each application of the methodology, the first sliding window is considered as the baseline period. Asterisks indicate values lying outside the plotted range.

The flattening of the probability curves when considering the changes of dependence only results in reducing the contribution of the dependence properties (Figs. S16c, f, i and l). It illustrates here that contribution results not only depend on the choice of the baseline period (as already discussed in sub-section 6.2 – Conclusion, discussion and future work / Discussion and perspectives –, L554-L564 of the initial article), but also on the choice of the window length of the baseline period.

As mentioned by the referee, 30 years can be limited to robustly study compound events and larger window length could be preferable. However, in Time of Emergence studies, i.e., in a climate change context, choosing a large window length may provide less informative results on the detection of emergence. Thus, to test and illustrate our methodology, we chose to use a 30-year sliding window, which can be seen as a trade-off between providing informative ToE results, and at the same time, having a sufficient sample size to robustly evaluate compound event probabilities.

However, in order to inform readers that ToE and contribution results can be modified by the choice of the window length, we have added the new Figures S15 and S16 in the supplementary materials, as well as the following modifications and sentences (in blue) in the sub-section 6.2 (Conclusion, discussion and future work / Discussion and perspectives) of the initially submitted article:

"*In this study, emergence of probabilities of multivariate hazards has been investigated with respect to the **30-year** baseline period 1871-1900. This period can be considered as representative of the beginning of the industrial era (e.g., Hawkins et al., 2020) and can hence be of interest to assess if anthropogenic climate change has contributed to an emergence of probability of multivariate hazards. However, other baseline periods could have been chosen, such as more recent ones which would provide useful results for adaptation planning (e.g., Ossó et al., 2022). Of course, depending on the chosen baseline period, the estimated natural variability that serves as reference for assessing changes would be different, and thus would affect the ToE results. As an illustration, Fig. S14 shows results from a quick sensitivity experiment for the time of emergence of probabilities of compounding wind and precipitation depending on the choice of the **30-year** baseline period for the CNRM-CM6 model. It illustrates that results of emergence can vary strongly depending on the chosen baseline period. In addition to modifying the potential time of emergence, the choice of the baseline period can also influence the results of contributions from the statistical properties changes (not shown), as these statistical changes are also assessed with respect to the baseline period. **ToE and contribution results could also be modified by the choice of the length of the sliding windows. For example, considering a larger window length could attenuate the changes of statistical properties between the baseline period and the sliding windows, thus modifying the ToE results. Also, in a transient climate context, this results in mixing different climate conditions and, thus, different statistical properties.***

*As an illustration, ToE and contribution results for the CNRM-CM6 simulations are presented in Figs. S15 and S16 of the Supplement by considering sliding windows of 40 years (baseline period: 1971-1910), 50 years (baseline period: 1971-1920) and 60 years (baseline period: 1971-1930) but are not commented on in the present study.*"

**Comment:**

Fig. 6: What is the effect of sample size on the shown patterns? More extreme values are more uncertain, and bivariate exceedances are more uncertain than univariate one, hence ToE should be shifted back in time. Interesting that one gets a generally relatively rich, non-trivial structure here.

**Response:**

Fig. RC1 (see below) shows the ToE results for varying exceedance thresholds for the CNRM-CM6 simulations by considering 30-year, 40-year, 50-year and 60-year sliding windows. Please note that the color scale has been changed from Fig. 6 to be consistent with the following comment. By comparing panels RC1a-c, d-f, g-i and j-l together, we can see that, depending on the window length, different results are obtained for ToE matrices. ToE values and their relationships with the sample size is not trivial. Panels RC1a-c d-f and g-i show the same patterns for ToE, with ToE being, indeed, shifted back in time for most of the bivariate thresholds. In particular, when using a 50-year sliding window, none of the bivariate thresholds present a probability emerging when considering the changes of dependence only (RC1i). ToE values being shifted back in time can be explained by a reduced estimated confidence interval for natural variability due to the increased sample size. However, increasing sample size can also lead to obtaining later ToE values: for 60-year sliding windows (panels RC1j-k), later ToE results are obtained compared to those obtained with 50-year sliding windows (RC1g-i). This is mainly due to 1) the definition of ToE we used, with ToE detection when the probability signal permanently exceeds a certain threshold, and 2) probability signals that are flattened by increasing the window length, as already explained for Fig. S15. By being flattened, probability signals could emerge later from the confidence interval even though the range of the confidence interval for natural variability is reduced.

Fig. RC2 shows median contribution matrices of the marginal, dependence and interaction terms by considering 30-year, 40-year, 50-year and 60-year sliding windows. Again, by comparing panels RC2a-c, d-f, g-i and j-l together, we can see that, depending on the window length, different contribution results are obtained. In particular, similar patterns for the contribution of marginal and dependence properties are obtained for 30-year and 40-year sliding windows (RC2a-b and d-e), with dependence changes playing an important role for the probability of high bivariate extreme events (upper-right area of the subplots). This area of exceedance thresholds for which dependence properties contribute greatly to probability changes is however smaller with 40 than with 30-year sliding

windows. The decrease of the importance of the dependence properties for probability of high bivariate extreme events is then confirmed when increasing the window length (RC2h, k). It results in having marginal properties mainly driving probability changes for all pairs of thresholds (Figs. RC2g, j). Concerning the interaction term (RC2c, f, i, l), contribution values are approximately equal to 0, highlighting again the negligible role of this term in probability changes regardless of the choice of the length of the sliding window.

Although these results are interesting, including them, either in the body of the study or in the Supplement, would make the paper more cumbersome, which we think is not appropriate. Please note that, in our response to the previous comment, we have already suggested adding new sentences in the sub-section 6.2 (Conclusion, discussion and future work/ Discussion and perspectives) to discuss window length and its effect on ToE and contribution results. We thus propose not to add further details for this point. However, investigating those patterns, the importance of the dependence on the probabilities of extremes, and how they are represented by climate models, is of course an interesting perspective to explore in future research.

[Figure]

Figure RC1: CNRM-CM6 time of emergence (at 68% confidence level) for compound wind and precipitation extremes due to changes of (a, d, g, j) both marginal and dependence properties, (b, e, h, k) marginal properties only, and (c, f, i, l) dependence properties only. Results are displayed for probabilities computed by using (a-c) 30-year, (d-f) 40-year, (g-i) 50-year and (j-l) 60-year windows sliding over the period 1871-2100. In each application of the methodology, the first sliding window is considered as the baseline period. White cells indicate that no time of emergence is detected, while white cells with red points indicate ToE values before 2020. Results are presented for varying exceedance thresholds between the 5th and 95th percentile of compound wind and precipitation extremes data.

[Figure]

Figure RC2: Matrices of median contributions of the (a, d, g, j) marginal, (b, e, h, k) dependence and (c, f, i, l) interaction terms. Results are displayed for probabilities computed by using (a-c) 30-year, (d-f) 40-year, (g-i) 50-year and (j-l) 60-year windows sliding over the period 1871-2100. In each application of the methodology, the first sliding window is considered as the baseline period. Results are presented for varying exceedance thresholds between the 5th and 95th percentile of compound wind and precipitation extremes data. Upper triangles show a contribution ≥ 50 %.

**Comment:**

For all figures: the colour scales for the years are not very intuitive. One continuous colour would make more sense.

**Response:**

We propose to change the color scales for the years for all figures (i.e., changing the color scale "Red-Yellow-Blue" to the color scale "Red-Yellow"). Please also note that, now, we are also  plotting only one color bar for each row and broadening them. As an illustration, we show here the modifications made for Fig. 6.

Previous Fig. 6:

[Figure]

Proposed new Fig. 6:

[Figure]

**Comment:**

Section 6 should be entitled: "Discussion and conclusion" or similar and then maybe "Summary" in Section 6.1.

**Response:**

We agree. Section 6 is now entitled "Summary and discussion" instead of "Conclusion, discussion and future work". Section 6.1 is now entitled "Summary".

---

## Author Comment (AC2)

**Response to Referee Comment 2 on "Time of Emergence of compound events: contribution of univariate and dependence properties" by Bastien François et al.**

**Anonymous Referee #2**

**General comments:**

**Comment:**

The authors present a nice framework for estimating the time of emergence (TOE) of compound events (CEs). To demonstrate the framework, the authors analyse the TOE for two types of CEs in France. While I recommend publication of the framework in general, I have two major concerns regarding the application which should be considered prior to acceptance.

**Response:**

We would like to thank the anonymous referee for her/his very positive comments and the detailed questions. All the comments and our point-by-point responses are given below.

**Comment:**

Major concerns:

1. The authors apply their framework to CMIP6 model data. While this choice may be justified for the temperature related CE in central France, it is likely not for the wind-precipitation CE in Brittanny. CMIP6 models have a rather coarse resolution (mostly coarser than the chosen grid of 0.5° x 0.5°). Their representation of regional wind and precipitation is therefore likely rather bad (see e.g. IPCC AR6 WG1 Chapter 10), and more representative of large scale averages which might not be relevant for regional impacts. Thus the value of the study is more on the development (and demonstration) of the framework rather than on providing relevant results for the two considered regions (maybe one of them).

**Response:**

We agree with this comment: the coarse resolution of CMIP6 models can affect the representation of regional compound events such as compounding wind and precipitation extremes over the region of Brittany, France. However, and as the referee noted, the aim of the paper is more on the development of the framework than on providing precise results for Time of Emergence of specific compound events. With the aim of clarifying better this point, we propose to add the following sentence (in blue) in the paragraph starting at L70 of the initial submitted article (Section Introduction):

> "*In this paper, we propose a new methodology to assess the time of emergence of compound events probabilities. We also develop a*

*copula-based multivariate framework, which allows for an adequate description of the contribution of the marginal and dependence properties changes to the evolutions of multivariate hazard probabilities. This compound event analysis is applied to two case studies.* ***Please note that the goal of the paper here is not to provide precise results of ToE on these two case studies, but rather to introduce the conceptual framework and raise awareness among climate scientists on the potential emergence of CE probabilities, as well as the contributions of statistical properties to probability changes.*** *We first analyse compound wind and precipitation extremes over the coastal region of Brittany (France). This bivariate compound event, i.e., composed of co-occurring climate hazards over the same region and time, has been analysed in several studies (e.g., Martius et al., 2016; Bevacqua et al., 2019; De Luca et al., 2020a; Reinert et al., 2021; Messmer and Simmonds, 2021) as it can have severe impacts such as important economic losses, massive damages to infrastructure and loss of human life (e.g., Fink et al., 2009; Liberato, 2014; Wahl et al., 2015; Raveh-Rubin and Wernli, 2015).*"

**Comment:**

Alternatively, the authors could have chosen CORDEX simulations. I guess they did not because they wanted to choose a preindustrial baseline. But is this really useful? People are (well, should be) adapted to present climate, so one could well estimate the TOE relative to, e.g., 1971-2000 or even a later period. This would also increase the relevance of the results for the chosen journal by essentially asking "when will we feel climate change in these hazards?". I therefore request to replace the chosen models by CORDEX models to provide results of practical relevance. I explicitly leave the decision on this choice to the editor as the paper anyway makes a useful conceptual contribution.

**Response:**

It is true that the choice of using simulations from CMIP6 models instead of CORDEX is in part due to our willingness to investigate probability changes with respect to a preindustrial baseline. This choice is also made in order to complement the research on the uncertainty of GCMs in representing the characteristics of CEs and their evolution over time (e.g., Ridder et al., 2021, 2022; Vogel et al., 2020, among others). Although we agree with the anonymous referee that analyzing outputs from RCMs with higher resolution (e.g., CORDEX simulations) could provide results of practical relevance at  regional scale, we believe that our study using GCM simulations still provides interesting results that illustrate our methodology well and highlight general key points when analyzing changes of CE probabilities. However, in order to clarify that our methodology can be used and/or adapted to analyse simulations with higher spatial resolution, we propose to add the following paragraph (in blue) in the sub-section 6.2 (Conclusion, discussion and future work / Discussion and perspectives) at L585 of the initial submitted article:

*"In this study, we demonstrated our conceptual framework using simulations from an ensemble of 13 GCMs. While using GCMs permitted to illustrate our methodology and draw general conclusions when analyzing changes of CE probabilities, the resolution of such climate models is often considered too coarse for a realistic representation of climate variables at a regional scale, such as for precipitation and wind (e.g., IPCC, 2021). Consequently, the ToE results obtained in this study for Brittany and Central France regions may not be accurate enough to be used for adaptation planning. Applying our methodology to analyse simulations from Regional Climate Models (RCMs) that simulate physical processes at a higher spatial resolution would permit us to provide more relevant information on regional CEs that could be used to design realistic regional adaptation strategies. An appropriate future step could be for example to apply the presented methodology to analyse CEs using simulations from RCMs forced by CMIP5 data (CORDEX) or CMIP6 data."*

Regarding the baseline period, we believe that the choice of a pre-industrial baseline period in our study permits us to provide interesting results on probabilities associated with natural variability and the influence of anthropogenic climate change on CEs. However, it is true that a more recent baseline period such as 1971-2000 could have been chosen to provide information adapted to the present (or say recent) climate. This is already mentioned in sub-section 3.1 (Statistical method/Time of emergence of climate hazards) at L160 of the initial submitted article:

*"In this study, we consider the reference period (1871-1900) as baseline to assess the emergence of hazard probabilities. However, there is no agreement on the choice of the baseline period for ToE studies. While most of the studies choose a pre-industrial period as baseline to attribute emergence to anthropogenic greenhouse gas forcing (e.g., 1850-1900, Hawkins et al., 2020), other studies choose a more recent baseline period (e.g., 1951-1983, Ossó et al., 2022), which can provide relevant information for adaptation planning. We further discuss the choice of the reference period for emergence in Sect. 6."*

and in sub-section 6.2 (Conclusion, discussion and future work / Discussion and perspectives) at L557:

*"However, other baseline periods could have been chosen, such as more recent ones which would provide useful results for adaptation planning (e.g., Ossó et al., 2022)."*

We however suggest to provide more information on the motivation for choosing more recent baseline periods by adding new sentences (in blue) in sub-section 6.2 (Conclusion, discussion and future work / Discussion and perspectives) at L557:

*"However, other baseline periods could have been chosen, such as more recent ones which would provide useful results for adaptation planning (e.g., Ossó et al., 2022). **In practice, despite the climate changes with respect to its natural variability, societies are (or should be) adapted to the present or recent climate. Hence, estimating the ToE relative to a more recent period, e.g. 1971-2000, would permit providing results of more practical relevance for adaptation.** Of course, depending on the chosen baseline period, the estimated natural variability that serves as reference for assessing changes would be different, and thus would affect the ToE results."*

**Comment:**

2. The authors make use of the model ensemble in two different ways (Indiv vs. Full).

The TOE compares noise and signal and is thus a property of the climate system. The noise should really just be internal variability, and the signal the forced signal, which differs from model to model. Pooling all models together mixes signal and noise - the difference in model signals is mixed into the noise. This makes no sense in particular because the uncertainties in precipitation and wind changes are so uncertain that calculating a mean change as signal may create something physically implausible (see e.g. Shepherd, Nat. Geosci, 2014). Please remove the "Full Ensemble" version!

**Response:**

We agree that pooling may lead to a loss of signal. The "Full-Ensemble" version proposes a methodology to extract a CE probability signal among the different models by assembling the variables contributing to the CE using pooling. The underlying philosophy of this procedure is similar to multi-model means (MMM, see, e.g., Tebaldi and Knutti 2007; Knutti et al. 2010) that consist in assembling the different models and averaging them to derive multi-model means. Pooling of model data has already been performed in the climate science literature and assumes that each climate model is one valid representation of the physical processes at play in the climate system, and climate simulations being samples from the observed distribution  (e.g., Srivastava et al., 2021). Under this assumption, it permits us to assess the global uncertainty inherent in climate modelling. Note also that a bias correction method has been applied to each climate simulation (with respect to CNRM-CM6) in order to make the simulations consistent with each other. However, it is true that, as for any MMM or pooling, in practice, model' signals are different and thus mixed together. Even if the application of the Full-version has some drawbacks that can be criticized, in particular for its application on wind and precipitation subject to large uncertainties as mentioned by the anonymous referee, we believe that the methodology and the results obtained are still interesting and can be of interest to readers investigating compound events probability using multi-model ensembles.

Hence, instead of completely removing the "Full-ensemble version" from the article, we propose to remove all explanations, results, figures and appendices related to the Full-version from the main body of the article, and put them as Supplementary material. In addition, we propose to also mention its limitations for variables with large uncertainty by adding a sentence (in blue) at the end of the paragraph starting at L273 of the initial submitted article for the "Full-Ensemble version" in the Supplement:

> "*Depending on the versions, the objectives are not the same: whereas the Indiv-Ensemble version permits to analyse the modelling of hazards separately and assess the uncertainty in ToE arising from the inter-model differences, the Full-Ensemble version permits to derive unique ToE estimates and contribution values accounting for the global uncertainty in climate modelling. This Full-Ensemble version assumes that the variables of interest are drawn from the same distribution.* **Please note that pooling multiple models together may lead to a loss of signal: by assembling the contributing variables together to estimate the CE probability, the pooling procedure can result in mixing the different model signals and noise together. This is particularly true for precipitation and wind changes which are subject to large uncertainties (Shepherd, 2014). However, pooling of model data assumes that each climate model is one valid representation of the physical processes at play in the climate system, and climate simulations being samples from the observed distribution (Srivastava et al., 2021). Thus, even if the climate models have different signals, pooling them and analysing them using our methodology with the Full-version permits to provide useful information on ToE and contribution values by taking into account the complete uncertainty in climate modelling.**"

Please note that, in agreement with the relocation of the results related to the Full-version in the Supplement, some of the figures in the article will be modified/reorganized. We here detail some of the main changes below. We stress that those changes are just a reorganization of the presentation of the results. It does not involve new figures and results.

- Figure 3: the right side of the flowchart concerning the Full-version will be deleted. A new figure for the flowchart concerning the Full-version will be added in the Supplement.
- Figure 7: Figs. 7a-c will be combined with Fig. S5a-c. A few sentences will be added to describe Fig. S5a-c in the main article..
- Figs. 7d-f will be combined with Figs. S5d-f in the Supplement.
- Figure 8: Bars corresponding to the Full-Ensemble version will be deleted. The resulting plot will be added in a panel in Fig.7 (panel Fig. 7g).
- Figures 9 and 10: Figs. 9d-f and Figs. 10d-f will be relocated in the Supplement into one figure. Figs. 9a-c and Figs. 10a-c will be combined into one new figure in the main article.
- Figure 11: Figure 11 will be deleted.
- Figure 12: Figs. 12a-c will be combined with Figs. S13a-c. A few sentences will be added to describe Figs. S13a-c in the main article.

- Figs. 12d-f will be combined with Figs. S13d-f in the Supplement.
- Figure 13: Bars corresponding to the Full-Ensemble version will be deleted. The resulting plot will be added in a panel in Fig. 12 (panel Fig. 12g)
- Figure S5: As explained, combine 7d-f with S5d-f.
- Figure S7 will be deleted.

**Comment:**

Minor issues:

line 46 and following:  More relevant in this context is detection rather than attribution. This should also be mentioned.

**Response:**

Following this comment, we propose the following corrections (in red and blue), L46 (Section Introduction):

> "*Evaluating the ToE of compound hazards probabilities with respect to a baseline period — from which the natural variability is estimated — is valuable to analyse evolutions of compound events and attribute those to a specific cause, such as anthropogenic greenhouse gas emissions.* **Detection and a***ttribution is an important research field in climate science that aims at determining the mechanisms responsible for recent global warming and related climate changes* **(Bindoff et al., 2013)***. For example, it can be done by comparing probabilities of an event between two worlds with different forcings (the "risk-based" approach, Stott et al., 2004; Shepherd, 2016).*"

**Comment:**

line 61: please refer to meteorological drought (no rain). You would likely find a change in soil moisture drought (low soil moisture).

**Response:**

We agree. We propose the following corrections (in blue), L61 (Section Introduction):

> "*For example, rising temperatures can naturally lead to more co-occurrences of hot temperatures and* **meteorological** *droughts, despite no significant trends in* **meteorological** *droughts are detected (Diffenbaugh et al., 2015; Mazdiyasni and AghaKouchak, 2015). However, in addition to warmer temperatures, the strengthening of the dependence between hot temperatures and* **meteorological** *droughts for future periods can also contribute to an increase in their co-occurrences (as highlighted in Zscheischler and Seneviratne, 2017).*"

**Comment:**

line 65 and following: This is not a general finding - it should be made clear for which type of CE this applies.

**Response:**

We propose to add the following corrections (in red and blue) starting at L64 of the initial article (Section Introduction):

> "*Several studies concluded about the importance of considering dependencies to assess CE properties and frequencies in a robust way**, e.g., for wind and precipitation extremes** ( Hillier et al., 2020**), or temperature and precipitation (** Singh et al., 2021a; Vrac et al., 2021). Recently, Abatzoglou et al. (2020) even showed **that, in the recent decades, changes in multivariate annual climatic conditions (water deficit, evapotranspiration, minimum and maximum temperature) with respect to a reference climate state have been more important than changes in univariate annual climatic conditions for a large portion of the Earth.**"*

**Comment:**

line 161: It is not really an issue that there is no agreement. In fact, there cannot be a single baseline as there is no single baseline climate. The point is that you simply ask a slightly different question when choosing a different baseline (e.g. compared to preindustrial; 1950s, 1990s, or even little ice age or medieval warm period). Also, as argued above, the TOE wrt to present climate might even be more relevant. I would rephrase this sentence or just delete it and simply state "We choose...".

**Response:**

We propose to delete the sentence (in red) starting L161 of the initially submitted article:

> "*In this study, we consider the reference period (1871-1900) as baseline to assess the emergence of hazard probabilities.  While most of the studies choose a pre-industrial period as baseline to attribute emergence to anthropogenic greenhouse gas forcing (e.g., 1850-1900, Hawkins et al., 2020), other studies choose a more recent baseline period (e.g., 1951-1983, Ossó et al., 2022), which can provide relevant information for adaptation planning.*"

**Comment:**

Beginning of Section 3.3: you should clarify that the attribution of changes to changes in margins and dependence has already been introduced by Bevacqua

et al., Sci. Adv. 2019. This also helps you to clarify the novel aspects of your contribution.

**Response:**

We agree and propose to add the following sentences (in blue) in the paragraph starting at L207 of the initial submitted article (Section 3.3 - Statistical method/Change of probabilities: contribution of the marginal and dependence properties):

> "*Then, do exceedance probability values change significantly between reference and future periods? And if so, how much of this change is due to changing marginal properties? To changing dependence structure?* **Attributing probability changes to changes of marginal and dependence properties has already been introduced by Bevacqua et al. (2019) to analyse compound flooding from precipitation and storm surge in Europe. However, to our knowledge, assessing those changes relative to a reference natural variability in a ToE context has never been done yet**. *In order to isolate the effects of these potentially changing statistical properties, we propose to calculate two additional exceedance probability values.*"

**Comment:**

Figure 6: the color bar is hardly visible. Please plot just one for each row and then broaden!

**Response:**

We agree on this comment not only for Fig. 6 but also for all the other figures, when applicable. As illustration, we show below the changes made for Fig. 6. Please note that we also considered the modification of the colour scale for ToE values.

Previous Fig. 6:

[Figure]

Proposed new Fig. 6:

[Figure]

**References:**

Bevacqua, E., Maraun, D., Vousdoukas, M. I., Voukouvalas, E., Vrac, M., Mentaschi, L., and Widmann, M.: Higher probability of compound flooding from precipitation and storm surge in Europe under anthropogenic climate change, Sci. Adv., 5, https://doi.org/10.1126/sciadv.aaw5531, 2019.

Bindoff N, Stott P, AchutaRao K, Allen M, Gillett N, D Gutzler D, K Hansingo K, Hegerl G, Hu Y, Jain S, Mokhov I, Overland J, Perlwitz J, Sebbari R, Zhang X (2013) Detection and attribution of climate change: from global to regional. In: Stocker TF, Qin D, Plattner G-K, Tignor M, Allen SK, Boschung J, Nauels A, Xia Y, Bex V, Midgley PM (eds) Climate change 2013: the physical science basis. Contribution of working group I to the fifth assessment report of the intergovernmental panel on climate change. Cambridge University Press, Cambridge

Christensen, J. H., Boberg, F., Christensen, O. B., and Lucas-Picher, P.: On the need for bias correction of regional climate change projections of temperature and precipitation, Geophys. Res. Lett., 35, L20 709, https://doi.org/10.1029/2008GL035694, 2008.

Knutti R, Furrer R, Tebaldi C, Cermak J, Meehl GA (2010) Challenges in combining projections from multiple climate models. J Clim 23(10):2739–2758. https://doi.org/10.1175/2009JCLI3361.1

Ridder, N., Pitman, A., and Ukkola, A.: Do CMIP6 Climate Models simulate Global or Regional Compound Events skilfully?, Geophys. Res. Lett., 48, https://doi.org/10.1029/2020GL091152, 2021.

Ridder, N., Ukkola, A., Pitman, A., and Perkins-Kirkpatrick, S.: Increased occurrence of high impact compound events under climate change, NPJ Clim. Atmos. Sci., 5, https://doi.org/10.1038/s41612-021-00224-4, 2022.

Srivastava, A. K., Grotjahn, R., Ullrich, P. A., and Sadegh, M.: Pooling Data Improves Multimodel IDF Estimates over Median-Based IDF Estimates: Analysis over the Susquehanna and Florida, J. Hydrometeorol., 22, 971–995, 2021.

Tebaldi C, Knutti R (2007) The use of the multi-model ensemble in probabilistic climate projections. Philos Trans R Soc A Math Phys Eng Sci 365(1857):2053–2075. https://doi.org/10.1098/rsta.2007.

Vogel, M.N., Hauser, M., & Seneviratne, S.I. (2020). Projected changes in hot, dry and wet extreme events' clusters in CMIP6 multi-model ensemble, Environ. Res. Lett. 15, 094021. https://doi.org/10.1088/1748-9326/ab90a7